# Understanding Intrinsic Robustness Using Label Uncertainty

**Xiao Zhang**
Department of Computer Science
University of Virginia
shawn@virginia.edu

**David Evans**
Department of Computer Science
University of Virginia
evans@virginia.edu

## Abstract

A fundamental question in adversarial machine learning is whether a robust classifier exists for a given task. A line of research has made some progress towards this goal by studying the concentration of measure, but we argue standard concentration fails to fully characterize the intrinsic robustness of a classification problem since it ignores data labels which are essential to any classification task. Building on a novel definition of label uncertainty, we empirically demonstrate that error regions induced by state-of-the-art models tend to have much higher label uncertainty than randomly-selected subsets. This observation motivates us to adapt a concentration estimation algorithm to account for label uncertainty, resulting in more accurate intrinsic robustness measures for benchmark image classification problems.

## 1 Introduction

Since the initial reports of adversarial examples against deep neural networks (Szegedy et al., 2014; Goodfellow et al., 2015), many defensive mechanisms have been proposed aiming to enhance the robustness of machine learning classifiers. Most have failed, however, against stronger adaptive attacks (Athalye et al., 2018; Tramer et al., 2020). PGD-based adversarial training (Mądry et al., 2018) and its variants (Zhang et al., 2019; Carmon et al., 2019) are among the few heuristic defenses that have not been broken so far, but these methods still fail to produce satisfactorily robust classifiers, even for classification tasks on benchmark datasets like CIFAR-10. Motivated by the empirical hardness of adversarially-robust learning, a line of theoretical works (Gilmer et al., 2018; Fawzi et al., 2018; Mahloujifar et al., 2019a; Shafahi et al., 2019) have argued that adversarial examples are unavoidable. In particular, these works proved that as long as the input distributions are concentrated with respect to the perturbation metric, adversarially robust classifiers do not exist. Recently, Mahloujifar et al. (2019b) and Prescott et al. (2021) generalized these results by developing empirical methods for measuring the concentration of arbitrary input distributions to derive an intrinsic robustness limit. (Appendix A provides a more thorough discussion of related work.)

We argue that the standard concentration of measure problem, which was studied in all of the aforementioned works, is not sufficient to capture a realistic intrinsic robustness limit for a classification problem. In particular, the standard concentration function is defined as an inherent property regarding the input metric probability space that does not take account of the underlying label information. We argue that such label information is essential for any supervised learning problem, including adversarially robust classification, so must be incorporated into intrinsic robustness limits.

**Contributions.** We identify the insufficiency of the standard concentration of measure problem and demonstrate why it fails to capture a realistic intrinsic robustness limit (Section 3). Then, we introduce the notion of *label uncertainty* (Definition 4.1), which characterizes the average uncertainty of label assignments for an input region. We then incorporate label uncertainty in the standard concentration measure as an initial step towards a more realistic characterization of intrinsic robustness (Section 4). Experiments on the CIFAR-10 and CIFAR-10H (Peterson et al., 2019) datasets demonstrate that error regions induced by state-of-the-art classification models all have high label uncertainty (Section 6.1), which validates the proposed label uncertainty constrained concentration problem.

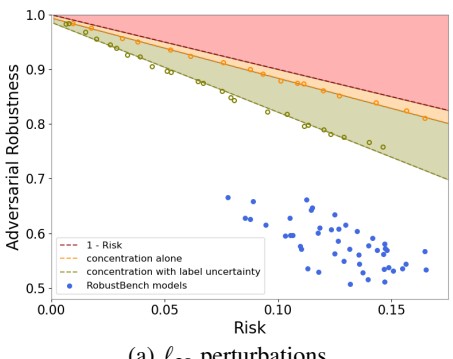 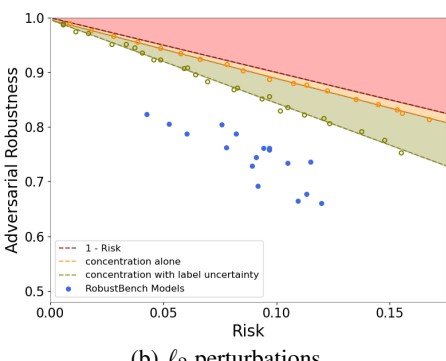

(a) $\ell_\infty$ perturbations        (b) $\ell_2$ perturbations

Figure 1: Intrinsic robustness estimates for classification tasks on CIFAR-10 under (a) $\ell_\infty$ perturbations with $\epsilon = 8/255$ and (b) $\ell_2$ perturbations with $\epsilon = 0.5$. Orange dots are intrinsic robustness estimates using the method in Prescott et al. (2021), which does not consider labels; green dots show the results using our methods that incorporate label uncertainty; blue dots are results achieved by the state-of-the-art adversarially-trained models in RobustBench (Croce et al., 2020). Three fundamental causes behind the adversarial vulnerability can be summarized as imperfect risk (red region), concentration of measure (orange region) and existence of uncertain inputs (green region).

By adapting the standard concentration estimation method in Mahloujifar et al. (2019b), we propose an empirical estimator for the label uncertainty constrained concentration function. We then theoretically study the asymptotic behavior of the proposed estimator and provide a corresponding heuristic algorithm for typical perturbation metrics (Section 5). We demonstrate that our method is able to produce a more accurate characterization of intrinsic robustness limit for benchmark datasets than was possible using prior methods that do not consider labels (Section 6.2). Figure 1 illustrates the intrinsic robustness estimates resulting from our label uncertainty approach on two CIFAR-10 robust classification tasks. The intrinsic robustness estimates we obtain by incorporating label uncertainty are much lower than prior limits, suggesting that compared with the concentration of measure phenomenon, the existence of uncertain inputs may explain more fundamentally the adversarial vulnerability of state-of-the-art robustly-trained models. In addition, we also provide empirical evidence showing that both the clean and robust accuracies of state-of-the-art robust classification models are largely affected by the label uncertainty of the tested examples, suggesting that adding an abstain option based on label uncertainty is a promising avenue for improving adversarial robustness of deployed machine learning systems (Section 6.3).

**Notation.** We use $[k]$ to denote $\{1, 2, \ldots, k\}$ and use $\mathbf{I}_n$ to denote the $n \times n$ identity matrix. For any set $\mathcal{A}$, $|\mathcal{A}|$ denotes its cardinality, $\text{pow}(\mathcal{A})$ is all its measurable subsets, and $\mathbb{1}_\mathcal{A}(\cdot)$ is the indicator function of $\mathcal{A}$. Consider metric probability space $(\mathcal{X}, \mu, \Delta)$, where $\Delta : \mathcal{X} \times \mathcal{X} \to \mathbb{R}_{\geq 0}$ is a distance metric on $\mathcal{X}$. Define the empirical measure of $\mu$ with respect to a data set $\mathcal{S}$ sampled from $\mu$ as $\widehat{\mu}_\mathcal{S}(\mathcal{A}) = \sum_{\boldsymbol{x} \in \mathcal{S}} \mathbb{1}_\mathcal{A}(\boldsymbol{x})/|\mathcal{S}|$. Denote by $\mathcal{B}_\epsilon(\boldsymbol{x}, \Delta)$ the ball around $\boldsymbol{x}$ with radius $\epsilon$ measured by $\Delta$. The $\epsilon$-expansion of $\mathcal{A}$ is defined as $\mathcal{A}_\epsilon(\Delta) = \{\boldsymbol{x} \in \mathcal{X} : \exists \, \boldsymbol{x}' \in \mathcal{B}_\epsilon(\boldsymbol{x}, \Delta) \cap \mathcal{A}\}$. When $\Delta$ is free of context, we simply write $\mathcal{B}_\epsilon(\boldsymbol{x}) = \mathcal{B}_\epsilon(\boldsymbol{x}, \Delta)$ and $\mathcal{A}_\epsilon = \mathcal{A}_\epsilon(\Delta)$.

## 2 Preliminaries

**Adversarial Risk.** Adversarial risk captures the vulnerability of a classifier against adversarial perturbations. In particular, we adopt the following adversarial risk definition, which has been studied in several previous works, such as Gilmer et al. (2018); Bubeck et al. (2019); Mahloujifar et al. (2019a;b); Zhang et al. (2020b); Prescott et al. (2021).

**Definition 2.1** (Adversarial Risk). Let $(\mathcal{X}, \mu, \Delta)$ be a metric probability space of instances and $\mathcal{Y}$ be the set of possible class labels. Assume $c : \mathcal{X} \to \mathcal{Y}$ is a concept function that gives each instance a label. For any classifier $f : \mathcal{X} \to \mathcal{Y}$ and $\epsilon \geq 0$, the *adversarial risk* of $f$ is defined as:

$$\text{AdvRisk}_\epsilon(f, c) = \Pr_{\boldsymbol{x} \sim \mu} \left[ \exists \, \boldsymbol{x}' \in \mathcal{B}_\epsilon(\boldsymbol{x}) \text{ s.t. } f(\boldsymbol{x}') \neq c(\boldsymbol{x}') \right].$$

The *adversarial robustness* of $f$ is defined as: $\text{AdvRob}_\epsilon(f, c) = 1 - \text{AdvRisk}_\epsilon(f, c)$.

When $\epsilon = 0$, adversarial risk equals to the standard risk. Namely, $\mathrm{AdvRisk}_0(f, c) = \mathrm{Risk}(f, c) := \mathrm{Pr}_{\boldsymbol{x} \sim \mu}[f(\boldsymbol{x}) \neq c(\boldsymbol{x})]$ holds for any classifier $f$. Other definitions of adversarial risk have been proposed, such as the one used in Mądry et al. (2018). These definitions are equivalent to the one we use, as long as small perturbations preserve the labels assigned by $c(\cdot)$.

**Intrinsic Robustness.** The definition of intrinsic robustness was first introduced by Mahloujifar et al. (2019b) to capture the maximum adversarial robustness with respect to some set of classifiers:

**Definition 2.2** (Intrinsic Robustness). Consider the input metric probability space $(\mathcal{X}, \mu, \Delta)$ and the set of labels $\mathcal{Y}$. Let $c : \mathcal{X} \to \mathcal{Y}$ be a concept function that gives a label to each input. For any set of classifiers $\mathcal{F} \subseteq \{f : \mathcal{X} \to \mathcal{Y}\}$ and $\epsilon \geq 0$, the *intrinsic robustness* with respect to $\mathcal{F}$ is defined as:

$$\overline{\mathrm{AdvRob}}_\epsilon(\mathcal{F}, c) = 1 - \inf_{f \in \mathcal{F}} \big\{ \mathrm{AdvRisk}_\epsilon(f, c) \big\} = \sup_{f \in \mathcal{F}} \{ \mathrm{AdvRob}_\epsilon(f, c) \}.$$

According to the definition of intrinsic robustness, there does not exist any classifier in $\mathcal{F}$ with adversarial robustness higher than $\overline{\mathrm{AdvRob}}_\epsilon(\mathcal{F}, c)$ for the considered task. Prior works, including Gilmer et al. (2018); Mahloujifar et al. (2019a;b); Zhang et al. (2020b), selected $\mathcal{F}$ in Definition 2.2 as the set of imperfect classifiers $\mathcal{F}_\alpha = \{f : \mathrm{Risk}(f, c) \geq \alpha\}$, where $\alpha \in (0, 1)$ is set as a small constant that reflects the best classification error rates achieved by state-of-the-art methods.

**Concentration of Measure.** Concentration of measure captures a 'closeness' property for a metric probability space of instances. More formally, it is defined by the concentration function:

**Definition 2.3** (Concentration Function). Let $(\mathcal{X}, \mu, \Delta)$ be a metric probability space. For any $\alpha \in (0, 1)$ and $\epsilon \geq 0$, *concentration function* is defined as: $h(\mu, \alpha, \epsilon) = \inf_{\mathcal{E} \in \mathsf{pow}(\mathcal{X})} \{\mu(\mathcal{E}_\epsilon) : \mu(\mathcal{E}) \geq \alpha\}$.

The standard notion of concentration function considers a special case of Definition 2.3 with $\alpha = 1/2$ (e.g., Talagrand (1995)). For some special metric probability spaces, one can prove the closed-form solution of the concentration function. The Gaussian Isoperimetric Inequality (Borell, 1975; Sudakov & Tsirelson, 1974) characterizes the concentration function for spherical Gaussian distribution and $\ell_2$-norm distance metric, and was generalized by Prescott et al. (2021) to other $\ell_p$ norms.

## 3 STANDARD CONCENTRATION IS INSUFFICIENT

We first explain a fundamental connection between the concentration of measure and the intrinsic robustness with respect to imperfect classifiers shown in previous work, and then argue that standard concentration fails to capture a realistic intrinsic robustness limit because it ignores data labels.

**Connecting Intrinsic Robustness with Concentration of Measure.** Let $(\mathcal{X}, \mu, \Delta)$ be the considered input metric probability space, $\mathcal{Y}$ be the set of possible labels, and $c : \mathcal{X} \to \mathcal{Y}$ be the concept function that gives each input a label. Given parameters $0 < \alpha < 1$ and $\epsilon \geq 0$, the standard concentration problem can be cast into an optimization problem as follows:

$$\underset{\mathcal{E} \in \mathsf{pow}(\mathcal{X})}{\text{minimize}} \ \mu(\mathcal{E}_\epsilon) \quad \text{subject to} \quad \mu(\mathcal{E}) \geq \alpha. \tag{3.1}$$

For any classifier $f$, let $\mathcal{E}_f = \{\boldsymbol{x} \in \mathcal{X} : f(\boldsymbol{x}) \neq c(\boldsymbol{x})\}$ be its induced error region with respect to $c(\cdot)$. By connecting the risk of $f$ with the measure of $\mathcal{E}_f$ and the adversarial risk of $f$ with the measure of the $\epsilon$-expansion of $\mathcal{E}_f$, Mahloujifar et al. (2019a) proved that the standard concentration problem (3.1) is equivalent to the following optimization problem regarding risk and adversarial risk:

$$\underset{f}{\text{minimize}} \ \mathrm{AdvRisk}_\epsilon(f, c) \quad \text{subject to} \quad \mathrm{Risk}(f, c) \geq \alpha.$$

To be more specific, the following lemma characterizes the connection between the standard concentration function and the intrinsic robustness limit with respect to the set of imperfect classifiers:

**Lemma 3.1** (Mahloujifar et al. (2019a)). Let $\alpha \in (0, 1)$ and $\mathcal{F}_\alpha = \{f : \mathrm{Risk}(f, c) \geq \alpha\}$ be the set of imperfect classifiers. For any $\epsilon \geq 0$, it holds that $\overline{\mathrm{AdvRob}}_\epsilon(\mathcal{F}_\alpha, c) = 1 - h(\mu, \alpha, \epsilon)$.

Lemma 3.1 suggests that the concentration function of the input metric probability space $h(\mu, \alpha, \epsilon)$ can be translated into an adversarial robustness upper bound that applies to any classifier with risk at

least $\alpha$. If this upper bound is shown to be small, then one can conclude that it is impossible to learn an adversarially robust classifier, as long as the learned classifier has risk at least $\alpha$.

**Concentration without Labels Mischaracterizes Intrinsic Robustness.** Despite the appealing relationship between concentration of measure and intrinsic robustness, we argue that solving the standard concentration problem is not enough to capture a meaningful intrinsic limit for adversarially robust classification. The standard concentration of measure problem (3.1), which aims to find the optimal subset that has the smallest $\epsilon$-expansion with regard to the input metric probability space $(\mathcal{X}, \mu, \Delta)$, does not involve the concept function $c(\cdot)$ that determines the underlying class label of each input. Therefore, no matter how we assign the labels to the inputs, the concentration function $h(\mu, \alpha, \epsilon)$ will remain the same for the considered metric probability space. In sharp contrast, learning an adversarially-robust classifier depends on the joint distribution of both the inputs and the labels.

Moreover, when the standard concentration function is translated into an intrinsic limit of adversarial robustness, it is defined with respect to the set of imperfect classifiers $\mathcal{F}_\alpha$ (see Lemma 3.1). The only restriction imposed by $\mathcal{F}_\alpha$ is that the classifier (or equivalently, the measure of the corresponding error region) has risk at least $\alpha$. This fails to consider whether the classifier is learnable or not under the given classification problem. Therefore, the intrinsic robustness limit implied by standard concentration $\overline{\mathrm{AdvRob}}_\epsilon(\mathcal{F}_\alpha, c)$ could be much higher than $\overline{\mathrm{AdvRob}}_\epsilon(\mathcal{F}_{\mathrm{learn}}, c)$, where $\mathcal{F}_{\mathrm{learn}}$ denotes the set of classifiers that can be produced by some supervised learning method. Hence, it is not surprising that Mahloujifar et al. (2019b) found that the adversarial robustness attained by state-of-the-art robust training methods for several image benchmarks is much lower than the intrinsic robustness limit implied by standard concentration of measure. In this work, to obtain a more meaningful intrinsic robustness limit we restrict the search space of the standard concentration problem (3.1) by considering both the underlying class labels and the learnability of the given classification problem.

**Gaussian Mixture Model.** We further illustrate the insufficiency of standard concentration under a simple Gaussian mixture model. Let $\mathcal{X} \subseteq \mathbb{R}^n$ be the input space and $\mathcal{Y} = \{-1, +1\}$ be the label space. Assume all the inputs are first generated according to a mixture of 2-Gaussian distribution: $\boldsymbol{x} \sim \mu = \frac{1}{2}\mathcal{N}(-\boldsymbol{\theta}, \sigma^2 \mathbf{I}_n) + \frac{1}{2}\mathcal{N}(\boldsymbol{\theta}, \sigma^2 \mathbf{I}_n)$, then labeled by a concept function $c(\boldsymbol{x}) = \mathrm{sgn}(\boldsymbol{\theta}^\top \boldsymbol{x})$, where $\boldsymbol{\theta} \in \mathbb{R}^n$ and $\sigma \in \mathbb{R}$ are given parameters (this concept function is also the Bayes optimal classifier, which best separates the two Gaussian clusters). Theorem 3.2, proven in Appendix C.1, characterizes the optimal solution to the standard concentration problem under this assumed model.

**Theorem 3.2.** Consider the above Gaussian mixture model with $\ell_2$ perturbation metric. The optimal solution to the standard concentration problem (3.1) is a halfspace, either

$$\mathcal{H}_- = \{\boldsymbol{x} \in \mathcal{X} : \boldsymbol{\theta}^\top \boldsymbol{x} + b \cdot \|\boldsymbol{\theta}\|_2 \leq 0\} \quad \text{or} \quad \mathcal{H}_+ = \{\boldsymbol{x} \in \mathcal{X} : \boldsymbol{\theta}^\top \boldsymbol{x} - b \cdot \|\boldsymbol{\theta}\|_2 \geq 0\},$$

where $b$ is a parameter depending on $\alpha$ and $\boldsymbol{\theta}$ such that $\mu(\mathcal{H}_-) = \mu(\mathcal{H}_+) = \alpha$.

**Remark 3.3.** Theorem 3.2 suggests that for the Gaussian mixture model, the optimal subset achieving the smallest $\epsilon$-expansion under $\ell_2$-norm distance metric is a halfspace $\mathcal{H}$, which is far away from the boundary between the two Gaussian classes for small $\alpha$. When translated into the intrinsic robustness problem, the corresponding optimal classifier $f$ has to be constructed by treating $\mathcal{H}$ as the only error region, or more precisely $f(\boldsymbol{x}) = c(\boldsymbol{x})$ if $\boldsymbol{x} \notin \mathcal{H}$; $f(\boldsymbol{x}) \neq c(\boldsymbol{x})$ otherwise. This optimally constructed classifier $f$, however, does not match our intuition of what a predictive classifier would do under the considered Gaussian mixture model. In particular, since all the inputs in $\mathcal{H}$ and their neighbours share the same class label and are also far away from the boundary, examples that fall into $\mathcal{H}$ should be easily classified correctly using simple decision rule, such as k-nearest neighbour or maximum margin, whereas examples that are close to the boundary should be more likely to be misclassified as errors by supervisedly-learned classifiers. This confirms our claim that standard concentration is not sufficient for capturing a meaningful intrinsic robustness limit.

## 4 Incorporating Label Uncertainty in Intrinsic Robustness

In this section, we first propose a new concentration estimation framework by imposing a constraint based on label uncertainty (Definition 4.1) on the search space with respect to the standard problem (3.1). Then, we explain why this yields a more realistic intrinsic robustness limit.

Let $(\mathcal{X}, \mu)$ be the input probability space and $\mathcal{Y} = \{1, 2, \ldots, k\}$ be the set of labels. $\eta : \mathcal{X} \to [0, 1]^k$ is said to capture the *full label distribution* (Geng, 2016; Gao et al., 2017), if $[\eta(\boldsymbol{x})]_y$ corresponds

to the description degree of $y$ to $\boldsymbol{x}$ for any $\boldsymbol{x} \in \mathcal{X}$ and $y \in \mathcal{Y}$, and $\sum_{y \in [k]} [\eta(\boldsymbol{x})]_y = 1$ holds for any $\boldsymbol{x} \in \mathcal{X}$. For classification tasks that rely on human labeling, one can approximate the label distribution for any input by collecting human labels from multiple human annotators. Our experiments use the CIFAR-10H dataset that did this for the CIFAR-10 test images (Peterson et al., 2019).

For any subset $\mathcal{E} \in \mathsf{pow}(\mathcal{X})$, we introduce *label uncertainty* to capture the average uncertainty level with respect to the label assignments of the inputs within $\mathcal{E}$:

**Definition 4.1** (Label Uncertainty). Let $(\mathcal{X}, \mu)$ be the input probability space and $\mathcal{Y} = \{1, 2, \ldots, k\}$ be the complete set of class labels. Suppose $c : \mathcal{X} \to \mathcal{Y}$ is a concept function that assigns each input $\boldsymbol{x}$ a label $y \in \mathcal{Y}$. Assume $\eta : \mathcal{X} \to [0, 1]^k$ is the underlying label distribution function, where $[\eta(\boldsymbol{x})]_y$ represents the description degree of $y$ to $\boldsymbol{x}$. For any subset $\mathcal{E} \in \mathsf{pow}(\mathcal{X})$ with measure $\mu(\mathcal{E}) > 0$, the *label uncertainty* (LU) of $\mathcal{E}$ with respect to $(\mathcal{X}, \mu)$, $c(\cdot)$ and $\eta(\cdot)$ is defined as:

$$\mathrm{LU}(\mathcal{E}; \mu, c, \eta) = \frac{1}{\mu(\mathcal{E})} \int_{\mathcal{E}} \left\{ 1 - [\eta(\boldsymbol{x})]_{c(\boldsymbol{x})} + \max_{y' \neq c(\boldsymbol{x})} [\eta(\boldsymbol{x})]_{y'} \right\} d\mu.$$

We define $\mathrm{LU}(\mathcal{E}; \mu, c, \eta)$ as the average label uncertainty for all the examples that fall into $\mathcal{E}$, where $1 - [\eta(\boldsymbol{x})]_{c(\boldsymbol{x})} + \max_{y' \neq c(\boldsymbol{x})} [\eta(\boldsymbol{x})]_{y'}$ represents the label uncertainty of a single example $\{\boldsymbol{x}, c(\boldsymbol{x})\}$. The range of label uncertainty is $[0, 2]$. For a single input, label uncertainty of $0$ suggests the assigned label fully captures the underlying label distribution; label uncertainty of $1$ means there are other classes as likely to be the ground-truth label as the assigned label; label uncertainty of $2$ means the input is mislabeled and there is a different label that represents the ground-truth label. Based on the notion of label uncertainty, we study the following constrained concentration problem:

$$\underset{\mathcal{E} \in \mathsf{pow}(\mathcal{X})}{\text{minimize}} \ \mu(\mathcal{E}_\epsilon) \quad \text{subject to} \quad \mu(\mathcal{E}) \geq \alpha \ \text{and} \ \mathrm{LU}(\mathcal{E}; \mu, c, \eta) \geq \gamma, \tag{4.1}$$

where $\gamma \in [0, 2]$ is a constant. When $\gamma$ is set as zero, (4.1) simplifies to the standard concentration of measure problem. In this work, we set the value of $\gamma$ to roughly represent the label uncertainty of the error region of state-of-the-art classifiers for the given classification problem.

Theorem 4.2, proven in Appendix C.2, shows how (4.1) captures the intrinsic robustness limit with respect to the set of imperfect classifiers whose error region label uncertainty is at least $\gamma$.

**Theorem 4.2.** Define $\mathcal{F}_{\alpha, \gamma} = \{f : \mathrm{Risk}(f, c) \geq \alpha, \mathrm{LU}(\mathcal{E}_f; \mu, c, \eta) \geq \gamma\}$, where $\alpha \in (0, 1)$, $\gamma \in (0, 2)$ and $\mathcal{E}_f = \{\boldsymbol{x} \in \mathcal{X} : f(\boldsymbol{x}) \neq c(\boldsymbol{x})\}$ is the error region of $f$. For any $\epsilon \geq 0$, it holds that

$$\inf_{\mathcal{E} \in \mathsf{pow}(\mathcal{X})} \{\mu(\mathcal{E}_\epsilon) : \mu(\mathcal{E}) \geq \alpha, \mathrm{LU}(\mathcal{E}; \mu, c, \eta) \geq \gamma\} = 1 - \overline{\mathrm{AdvRob}}_\epsilon(\mathcal{F}_{\alpha, \gamma}, c).$$

Compared with standard concentration, (4.1) aims to search for the least expansive subset with respect to input regions with high label uncertainty. According to Theorem 4.2, the translated intrinsic robustness limit is defined with respect to $\mathcal{F}_{\alpha, \gamma}$ and is guaranteed to be no greater than $\overline{\mathrm{AdvRob}}_\epsilon(\mathcal{F}_\alpha)$. Although both $\overline{\mathrm{AdvRob}}_\epsilon(\mathcal{F}_\alpha)$ and $\overline{\mathrm{AdvRob}}_\epsilon(\mathcal{F}_{\alpha, \gamma})$ can serve as valid robustness upper bounds for any $f \in \mathcal{F}_{\alpha, \gamma}$, the latter one would be able to capture a more meaningful intrinsic robustness limit, since state-of-the-art classifiers are expected to more frequently misclassify inputs with large label uncertainty, as there is more discrepancy between their assigned labels and the underlying label distribution (Section 6 provides supporting empirical evidence for this on CIFAR-10).

**Need for Soft Labels.** The proposed approach requires label uncertainty information for training examples. The CIFAR-10H dataset provided soft labels from humans that enabled our experiments, but typical machine learning datasets do not provide such information. Below, we discuss possible avenues to estimating label uncertainty when human soft labels are not available and are too expensive to acquire. A potential solution is to estimate the set of examples with high label uncertainty using the predicted probabilities of a classification model. Confident learning (Natarajan et al., 2013; Lipton et al., 2018; Huang et al., 2019; Northcutt et al., 2021b;a) provides a systematic method to identify label errors in a dataset based on this idea. If the estimated label errors match the examples with high human label uncertainty, then we can directly extend our framework by leveraging the estimated error set. Our experiments on CIFAR-10 (see Appendix G), however, suggest that there is a misalignment between human recognized errors and errors produced by confident learning. The existence of such misalignment further suggests that one should be cautious when combining the estimated set of label errors into our framework. As the field of confident learning advances to produce a more accurate estimator of label error set, it would serve as a good alternative solution for applying our framework to the setting where human label information is not accessible.

## 5 MEASURING CONCENTRATION WITH LABEL UNCERTAINTY CONSTRAINTS

Directly solving (4.1) requires the knowledge of the underlying input distribution $\mu$ and the ground-truth label distribution function $\eta(\cdot)$, which are usually not available for classification problems. Thus, we consider the following empirical counterpart of (4.1):

$$\underset{\mathcal{E} \in \mathcal{G}}{\text{minimize}} \ \widehat{\mu}_{\mathcal{S}}(\mathcal{E}_{\epsilon}) \quad \text{subject to} \quad \widehat{\mu}_{\mathcal{S}}(\mathcal{E}) \geq \alpha \ \text{and} \ \text{LU}(\mathcal{E}; \widehat{\mu}_{\mathcal{S}}, c, \widehat{\eta}) \geq \gamma, \tag{5.1}$$

where the search space is restricted to some specific collection of subsets $\mathcal{G} \subseteq \text{pow}(\mathcal{X})$, $\mu$ is replaced by the empirical distribution $\widehat{\mu}_{\mathcal{S}}$ with respect to a set of inputs sampled from $\mu$, and the empirical label distribution $\widehat{\eta}(\boldsymbol{x})$ is considered as an empirical replacement of $\eta(\boldsymbol{x})$ for any given input $\boldsymbol{x} \in \mathcal{S}$.

Theorem 5.1, proven in Appendix C.3, characterizes a generalization bound regarding the proposed label uncertainty estimate. It shows that if $\mathcal{G}$ is not too complex and $\widehat{\eta}$ is close to the ground-truth label distribution function $\eta$, the empirical estimate of label uncertainty $\text{LU}(\mathcal{E}; \widehat{\mu}_{\mathcal{S}}, c, \widehat{\eta})$ is guaranteed to be close to the actual label uncertainty $\text{LU}(\mathcal{E}; \mu, c, \eta)$. The formal definition of the complexity penalty with respect to a collection of subsets is given in Appendix B.

**Theorem 5.1** (Generalization of Label Uncertainty). Let $(\mathcal{X}, \mu)$ be a probability space and $\mathcal{G} \subseteq \text{pow}(\mathcal{X})$ be a collection of subsets of $\mathcal{X}$. Assume $\phi : \mathbb{N} \times \mathbb{R} \rightarrow [0, 1]$ is a complexity penalty for $\mathcal{G}$. If $\widehat{\eta}(\cdot)$ is close to $\eta(\cdot)$ in $L^1$-norm with respect to $\mu$, i.e. $\int_{\mathcal{X}} \|\eta(\boldsymbol{x}) - \widehat{\eta}(\boldsymbol{x})\|_1 d\mu \leq \delta_{\eta}$, where $\delta_{\eta} \in (0, 1)$ is a small constant, then for any $\alpha, \delta \in (0, 1)$ such that $\delta < \alpha$, we have

$$\Pr_{\mathcal{S} \leftarrow \mu^m} \left[ \exists \, \mathcal{E} \in \mathcal{G} \text{ and } \mu(\mathcal{E}) \geq \alpha : \left| \text{LU}(\mathcal{E}; \mu, c, \eta) - \text{LU}(\mathcal{E}; \widehat{\mu}_{\mathcal{S}}, c, \widehat{\eta}) \right| \leq \frac{4\delta + \delta_{\eta}}{\alpha - \delta} \right] \leq \phi(m, \delta).$$

**Remark 5.2.** Theorem 5.1 implies the generalization of concentration under label uncertainty constraints (see Theorem C.3 for a formal argument of this and its proof in Appendix C.5). If we choose $\mathcal{G}$ and the collection of its $\epsilon$-expansions, $\mathcal{G}_{\epsilon} = \{\mathcal{E}_{\epsilon} : \mathcal{E} \in \mathcal{G}\}$, in a careful way that both of their complexities are small, then with high probability, the empirical label uncertainty constrained concentration will be close to the actual concentration when the search space is restricted to $\mathcal{G}$.

Moreover, define $h(\mu, c, \eta, \alpha, \gamma, \epsilon, \mathcal{G}) = \inf_{\mathcal{E} \in \mathcal{G}} \{\mu(\mathcal{E}_{\epsilon}) : \mu(\mathcal{E}) \geq \alpha, \text{LU}(\mathcal{E}; \mu, c, \eta) \geq \gamma\}$ as the generalized concentration function under label uncertainty constraints. Then, based on a similar proof technique used for Theorem 3.5 in Mahloujifar et al. (2019b), we can further show that if $\mathcal{G}$ also satisfies a universal approximation property, then with probability 1,

$$h(\mu, c, \eta, \alpha, \gamma - \delta_{\eta}/\alpha, \epsilon) \leq \lim_{T \rightarrow \infty} h(\mu_{\mathcal{S}_T}, c, \widehat{\eta}, \alpha, \gamma, \epsilon, \mathcal{G}(T)) \leq h(\mu, c, \eta, \alpha, \gamma + \delta_{\eta}/\alpha, \epsilon), \tag{5.2}$$

where $T$ stands for the complexity of $\mathcal{G}$ and $\mathcal{S}_T$ denotes a set of samples of size $m(T)$. Appendix C.4 provides a formal argument and proof for (5.2). It is worth noting that (5.2) suggests that if we increase both the complexity of the collection of subsets $\mathcal{G}$ and the number of samples used for the empirical estimation, the optimal value of the empirical concentration problem (5.1) will converge to the actual concentration function with an error limit of $\delta_{\eta}/\alpha$ on parameter $\gamma$. When the difference between the empirical label distribution $\widehat{\eta}(\cdot)$ and the underlying label distribution $\eta(\cdot)$ is negligible, it is guaranteed that the optimal value of (5.1) asymptoptically converges to that of (4.1).

**Concentration Estimation Algorithm.** Although Remark 5.2 provides a general idea how to choose $\mathcal{G}$ for measuring concentration, it does not indicate how to solve the empirical concentration problem (5.1) for a specific perturbation metric. This section presents a heuristic algorithm for estimating the least-expansive subset for optimization problem (5.1) when the metric is $\ell_2$-norm or $\ell_{\infty}$-norm. We choose $\mathcal{G}$ as a union of balls for the $\ell_2$-norm distance metric and set $\mathcal{G}$ as a union of hypercubes for $\ell_{\infty}$-norm (see Appendix B for the formal definition of union of $\ell_p$-balls). It is worth noting that such choices of $\mathcal{G}$ satisfy the condition required for Theorem C.2, since they are universal approximators for any set and the VC-dimensions of both $\mathcal{G}$ and $\mathcal{G}_{\epsilon}$ are both bounded (see Eisenstat & Angluin (2007) and Devroye et al. (2013)).

The remaining task is to solve (5.1) based on the selected $\mathcal{G}$. Following Mahloujifar et al. (2019b), we place the balls for $\ell_2$ (or the hypercubes for $\ell_{\infty}$) in a sequential manner, and search for the best placement that satisfies the label uncertainty constraint using a greedy approach. Algorithm 1 in Appendix D gives pseudocode for the search algorithm. It initializes the feasible set of the hyperparmeters $\Omega$ as an empty set for each placement of balls (or hypercubes), then enumerates all

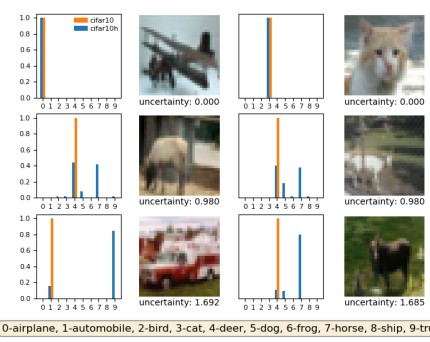 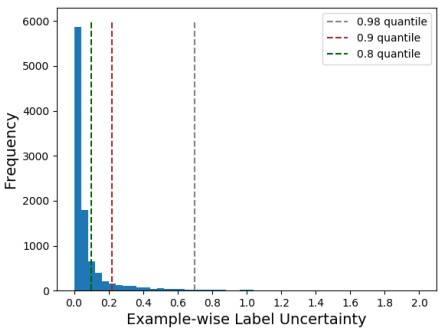

(a) Illustration of CIFAR-10 and CIFAR-10H          (b) Label Uncertainty Distribution

Figure 2: (a) Visualization of the CIFAR-10 test images with the soft labels from CIFAR-10H, the original assigned labels from CIFAR-10 and the label uncertainty scores computed based on Definition 4.1. (b) Histogram of the label uncertainty distribution for the CIFAR-10 test dataset.

the possible initial placements, $\mathcal{S}_{\text{init}}(\boldsymbol{u}, k)$, such that its empirical label uncertainty exceeds the given threshold $\gamma$. Finally, among all the feasible ball (or hypercube) placements, it records the one that has the smallest $\epsilon$-expansion with respect to the empirical measure $\widehat{\mu}_{\mathcal{S}}$. In this way, the input region produced by Algorithm 1 serves as a good approximate solution to the empirical problem (5.1).

## 6 EXPERIMENTS

We conduct experiments on the CIFAR-10H dataset (Peterson et al., 2019), which contains soft labels reflecting human perceptual uncertainty for the 10,000 CIFAR-10 test images (Krizhevsky & Hinton, 2009). These soft labels can be regarded as an approximation of the label distribution function $\eta(\cdot)$ at each given input, whereas the original CIFAR-10 test dataset provides the class labels given by the concept function $c(\cdot)$. We report on experiments showing the connection between label uncertainty and classification error rates (Section 6.1) and that incorporating label uncertainty enables better intrinsic robustness estimates (Section 6.2). Section 6.3 demonstrates the possibility of improving model robustness by abstaining for inputs in high label uncertainty regions.

### 6.1 ERROR REGIONS HAVE LARGER LABEL UNCERTAINTY

Figure 2(a) shows the label uncertainty scores for several images with both the soft labels from CIFAR-10H and the original class labels from CIFAR-10 (see Appendix F for more illustrations). Images with low uncertainty scores are typically easier for humans to recognize their class category (first row of Figure 2(a)), whereas images with high uncertainty scores look ambiguous or even misleading (second and third rows). Figure 2(b) shows the histogram of the label uncertainty distribution for all the $10,000$ CIFAR-10 test examples. In particular, more than $80\%$ of the examples have label uncertainty scores below $0.1$, suggesting the original class labels mostly capture the underlying label distribution well. However, around $2\%$ of the examples have label uncertainty scores exceeding $0.7$, and some $400$ images appear to be mislabeled with uncertainty scores above $1.2$.

We hypothesize that ambiguous or misleading images should also be more likely to be misclassified as errors by state-of-the-art machine learning classifiers. That is, their induced error regions should have larger that typical label uncertainty. To test this hypothesis, we conduct experiments on CIFAR-10 and CIFAR-10H datasets. More specifically, we train different classification models, including intermediate models extracted at different epochs, using the CIFAR-10 training dataset, then empirically compute the standard risk, adversarial risk, and label uncertainty of the corresponding error region. The results are shown in Figure 3 (see Appendix E for experimental details).

Figures 3(a) and 3(b) demonstrate the relationship between label uncertainty and standard risk for various classifiers produced by standard training and adversarial training methods under $\ell_\infty$ perturbations with $\epsilon = 8/255$. In addition, we plot the label uncertainty with error bars of randomly-selected images from the CIFAR-10 test dataset as a reference. As the model classification accuracy increases, the label uncertainty of its induced error region increases, suggesting the misclassified examples tend to have higher label uncertainty. This observation holds consistently for both standard and

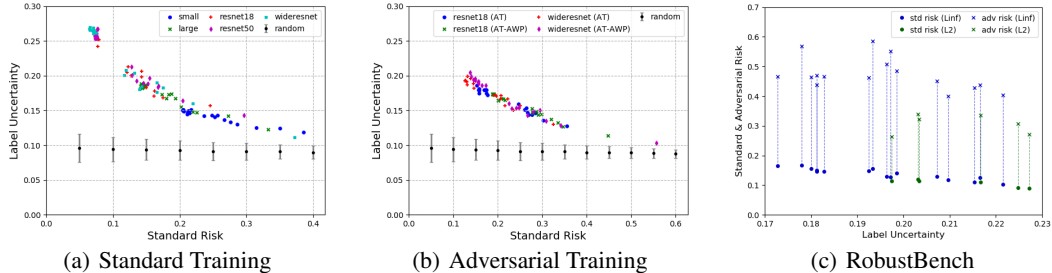

(a) Standard Training     (b) Adversarial Training     (c) RobustBench

Figure 3: Visualizations of error region label uncertainty versus standard risk and adversarial risk with respect to classifiers produced by different machine learning methods: (a) Standard-trained classifiers with different network architecture; (b) Adversarially-trained classifiers using different learning algorithms; (c) State-of-the-art adversarially robust classification models from RobustBench.

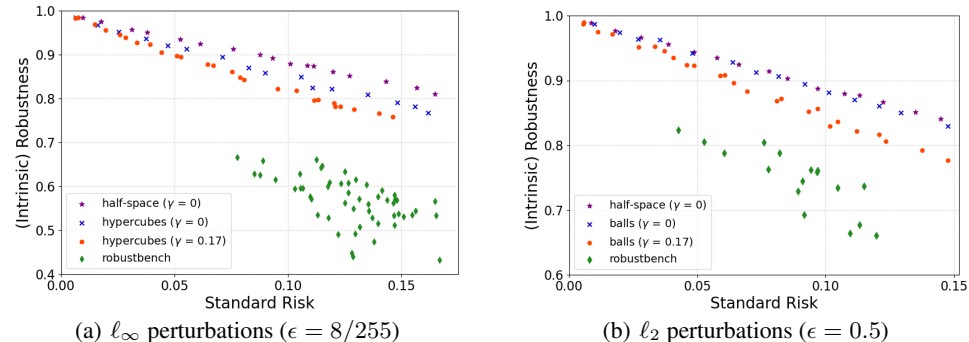

(a) $\ell_\infty$ perturbations ($\epsilon = 8/255$)     (b) $\ell_2$ perturbations ($\epsilon = 0.5$)

Figure 4: Estimated intrinsic robustness based on Algorithm 1 with $\gamma = 0.17$ under (a) $\ell_\infty$ perturbations with $\epsilon = 8/255$; and (b) $\ell_2$ perturbations with $\epsilon = 0.5$. For comparison, we plot baseline estimates produced without considering label uncertainty using a half-space searching method (Prescott et al., 2021) and using union of hypercubes or balls (Algorithm 1 with $\gamma = 0$). Robust accuracies achieved by state-of-the-art RobustBench models are plotted in green.

adversarially trained models with any tested network architecture. Figure 3(c) summarizes the error region label uncertainty with respect to the state-of-the-art adversarially robust models documented in RobustBench (Croce et al., 2020). Regardless of the perturbation type or the learning method, the average label uncertainty of their misclassified examples all falls into a range of $(0.17, 0.23)$, whereas the mean label uncertainty of all the testing CIFAR-10 data is less than $0.1$. This supports our hypothesis that error regions of state-of-the-art classifiers tend to have larger label uncertainty, and our claim that intrinsic robustness estimates should account for labels.

## 6.2 EMPIRICAL ESTIMATION OF INTRINSIC ROBUSTNESS

In this section, we apply Algorithm 1 to estimate the intrinsic robustness limit for the CIFAR-10 dataset under $\ell_\infty$ perturbations with $\epsilon = 8/255$ and $\ell_2$ perturbations with $\epsilon = 0.5$. We set the label uncertainty threshold $\gamma = 0.17$ to roughly represent the error region label uncertainty of state-of-the-art classification models (see Figure 3). In particular, we adopt a $50/50$ train-test split over the original $10,000$ CIFAR-10 test images (see Appendix E for experimental details).

Figure 4 shows our intrinsic robustness estimates with $\gamma = 0.17$ when choosing different values of $\alpha$. We include the estimates of intrinsic robustness defined with $\mathcal{F}_\alpha$ as a baseline, where no label uncertainty constraint is imposed ($\gamma = 0$). Results are shown both for our $\ell_p$-balls searching method and the half-space searching method in Prescott et al. (2021). We also plot the standard error and the robust accuracy of the state-of-the-art adversarially robust models in RobustBench (Croce et al., 2020). For concentration estimation methods, the plotted values are the empirical measure of the returned optimally-searched subset ($x$-axis) and $1-$ the empirical measure of its $\epsilon$-expansion ($y$-axis).

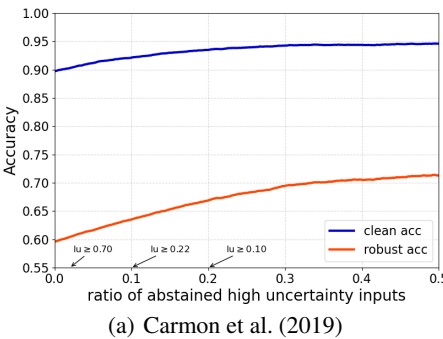 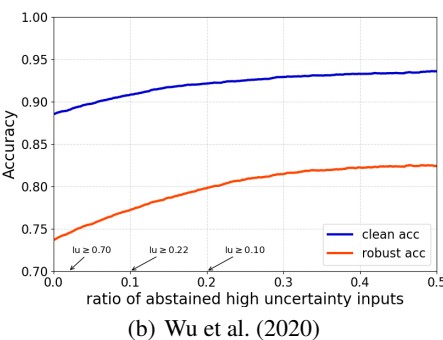

(a) Carmon et al. (2019)    (b) Wu et al. (2020)

Figure 5: Accuracy curves for different adversarially-trained classifiers, varying the abstaining ratio of CIFAR-10 images with high label uncertainty score: (a) Carmon et al. (2019) for $\ell_\infty$ perturbations with $\epsilon = 8/255$; (b) Wu et al. (2020) for $\ell_2$ perturbations with $\epsilon = 0.5$. Corresponding cut-off values of label uncertainty are marked on the $x$-axis with respect to percentage values of $\{0.02, 0.1, 0.2\}$.

Compared with the baseline estimates, our label-uncertainty constrained intrinsic robustness estimates are uniformly lower across all the considered settings (similar results are obtained under other experimental settings, see Table 1 in Appendix F). Although both of these estimates can serve as legitimate upper bounds on the maximum achievable adversarial robustness for the given task, our estimate, which takes data labels into account, being closer to the robust accuracy achieved by state-of-the-art classifiers indicates it is a more accurate characterization of intrinsic robustness limit. For instance, under $\ell_\infty$ perturbations with $\epsilon = 8/255$, the best adversarially-trained classification model achieves $66\%$ robust accuracy with approximately $8\%$ clean error, whereas our estimate indicates that the maximum robustness one can hope for is about $82\%$ as long as the classification model has at least $8\%$ clean error. In contrast, the intrinsic robustness limit implied by standard concentration is as high as $90\%$ for the same setting, which again shows the insufficiency of standard concentration.

### 6.3 Abstaining based on Label Uncertainty

Based on the definition of label uncertainty, and our experimental results in the previous subsections, we expect classification models to have higher accuracy on examples with low label uncertainty. Figure 5 shows the results of experiments to study the effect of abstaining based on label uncertainty on both clean and robust accuracies using adversarially-trained CIFAR-10 classification models from Carmon et al. (2019) ($\ell_\infty$, $\epsilon = 8/255$) and Wu et al. (2020) ($\ell_2$, $\epsilon = 0.5$). We first sort all the test CIFAR-10 images based on label uncertainty, then evaluate the model performance with respect to different abstaining ratios of top uncertain inputs. The accuracy curves suggest that a potential way to improve the robustness of classification systems is to enable the classifier an option to abstain on examples with high uncertainty score.

For example, if we allow the robust classifier of Carmon et al. (2019) to abstain on the 2% of the test examples whose label uncertainty exceeds 0.7, the clean accuracy improves from 89.7% to 90.3%, while the robust accuracy increases from 59.5% to 60.4%. This is close to the maximum robust accuracy that could be achieved with a 2% abstention rate ($0.595/(1 - 0.02) = 0.607$). This result points to abstaining on examples in high label uncertainty regions as a promising path towards achieving adversarial robustness.

## 7 Conclusion

Standard concentration fails to sufficiently capture intrinsic robustness since it ignores data labels. Based on the definition of label uncertainty, we observe that the error regions induced by state-of-the-art classification models all tend to have high label uncertainty. This motivates us to develop an empirical method to study the concentration behavior regarding the input regions with high label uncertainty, which results in more accurate intrinsic robustness measures for benchmark image classification tasks. Our experiments show the importance of considering labels in understanding intrinsic robustness, and further suggest that abstaining based on label uncertainty could be a potential method to improve the classifier accuracy and robustness.

## AVAILABILITY

An implementation of our method, and code for reproducing our experiments, is available under an open source license from: https://github.com/xiaozhanguva/intrinsic_rob_lu.

## ACKNOWLEDGEMENTS

This work was partially funded by an award from the National Science Foundation (NSF) SaTC program (Center for Trustworthy Machine Learning, #1804603).

## ETHICS STATEMENT

Our work is primarily focused on deepening our understanding of intrinsic adversarial robustness limit and the main contributions in this paper are theoretical. Our work could potentially enable construction of more robust classification systems, as suggested by the results in Section 6.3. For most applications, such as autonomous vehicles and malware detection, improving the robustness of classifiers is beneficial to society. There may be scenarios, however, such as face recognition where uncertainty and the opportunity to confuse classifiers with adversarial perturbations may be useful, so enabling more robust classifiers in these domains may have negative societal impacts.

## REPRODUCIBILITY STATEMENT

Details of our experimental setup and methods are provided in Appendix E, and all of the datasets we use are publicly available. In addition, we state the assumptions for our theoretical results in each theorem. Detailed proofs of all the presented theorems are provided in Appendix C.

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

## A  RELATED WORK

This section summarizes the work related to ours, beyond the brief background provided in the Introduction. First, we discuss the line of research aiming to develop robust classification models against adversarial examples. Then, we introduce the line of works which focus on understanding the intrinsic robustness limit.

## A.1 Training Adversarially Robust Classifiers

Witnessing the vulnerability of modern machine learning models to adversarial examples, extensive studies have been carried out aiming to build classification models that can be robust against adversarial perturbations. Heuristic defense mechanisms (Goodfellow et al., 2015; Papernot et al., 2016; Guo et al., 2018; Xie et al., 2018; Mądry et al., 2018) had been most popular until many of them were broken by stronger adaptive adversaries (Athalye et al., 2018; Tramer et al., 2020). The only scalable defense which seems to hold up well against adaptive adversaries is PGD-based adversarial training (Mądry et al., 2018). Several variants of PGD-based adversarial training have been proposed, which either adopt different loss function (Zhang et al., 2019; Wu et al., 2020) or make use of additional training data (Carmon et al., 2019; Alayrac et al., 2019). Nevertheless, the best current adversarially-trained classifiers can only achieve around $65\%$ robust accuracy on CIFAR-10 against $\ell_\infty$ perturbations with strength $\epsilon = 8/255$, even with additional training data (see the leaderboard in Croce et al. (2020)).

To end the arms race between heuristic defenses and newly designed adaptive attacks that break them, certified defenses have been developed based on different approaches, including linear programming (Wong & Kolter, 2018; Wong et al., 2018), semidefinite programming (Raghunathan et al., 2018), interval bound propagation (Gowal et al., 2019; Zhang et al., 2020a) and randomized smoothing (Cohen et al., 2019; Li et al., 2019). Although certified defenses are able to train classifiers with robustness guarantees for input instances, most defenses can only scale to small networks and they usually come with sacrificed empirical robustness, especially for larger adversarial perturbations.

## A.2 Theoretical Understanding on Intrinsic Robustness

Given the unsatisfactory status quo of building adversarially robust classification models, a line of research (Gilmer et al., 2018; Fawzi et al., 2018; Mahloujifar et al., 2019a; Shafahi et al., 2019; Dohmatob, 2019; Bhagoji et al., 2019) attempted to explain the adversarial vulnerability from a theoretical perspective. These works proved that as long as the input distribution is concentrated with respect to the perturbation metric, adversarially robust classifiers cannot exist. At the core of these results is the fundamental connection between the concentration of measure phenomenon and an intrinsic robustness limit that capture the maximum adversarial robustness with respect to some specific set of classifiers. For instance, Gilmer et al. (2018) showed that for inputs sampled from uniform $n$-spheres, a model-independent robustness upper bound under the Euclidean distance metric can be derived using the Gaussian Isoperimetric Inequality (Sudakov & Tsirelson, 1974; Borell, 1975). Mahloujifar et al. (2019a) generalized their result to any concentrated metric probability space of inputs. Nevertheless, it is unclear how to apply these theoretical results to typical image classification tasks, since whether or not natural image distributions are concentrated is unknown.

To address this question, Mahloujifar et al. (2019b) proposed a general way to empirically measure the concentration for any input distribution using data samples, then employed it to estimate an intrinsic robustness limit for typical image benchmarks. By showing the existence of a large gap between the limit implied by concentration and the empirical robustness achieved by state-of-the-art adversarial training methods, Mahloujifar et al. (2019b) further concluded that concentration of measure can only explain a small portion of adversarial vulnerability of existing image classifiers. More recently, Prescott et al. (2021) further strengthened their conclusion by using the set of half-spaces to estimate the concentration function, which achieves enhanced estimation accuracy. Other related works (Fawzi et al., 2018; Krusinga et al., 2019; Zhang et al., 2020b) proposed estimating lower bounds on the concentration of measure by approximating the underlying distribution using generative models. None of these works, however, consider data labels. Our main results show that data labels are essential for understanding intrinsic robustness limits.

## B Formal Definitions

In this section, we introduce the formal definitions of complexity penalty and union of $\ell_p$-balls that are used in Section 5. To begin with, we lay out the definition of complexity penalty that is defined for some collection of subsets $\mathcal{G} \in \mathsf{pow}(\mathcal{X})$. VC dimension and Rademacher complexity are commonly-used examples of such a complexity penalty.

**Definition B.1** (Complexity Penalty). Let $\mathcal{G} \subseteq \mathsf{pow}(\mathcal{X})$. We say $\phi : \mathbb{N} \times \mathbb{R} \to [0, 1]$ is a complexity penalty for $\mathcal{G}$, if for any $\delta \in (0, 1)$, it holds that

$$\Pr_{\mathcal{S} \leftarrow \mu^m} \left[ \exists \, \mathcal{E} \in \mathcal{G} : |\widehat{\mu}_{\mathcal{S}}(\mathcal{E}) - \mu(\mathcal{E})| \geq \delta \right] \leq \phi(m, \delta).$$

Next, we provide the formal definition of union of $\ell_p$-balls as follows:

**Definition B.2** (Union of $\ell_p$-Balls). Let $p \geq 1$. For any $T \in \mathbb{Z}^+$, define the union of $T$ $\ell_p$-balls as

$$\mathcal{B}(T; \ell_p) = \left\{ \cup_{t=1}^T \mathcal{B}_{\boldsymbol{r}_t}^{(\ell_p)}(\boldsymbol{u}_t) \colon \forall t \in [T], (\boldsymbol{u}_t, \boldsymbol{r}_t) \in \mathbb{R}^n \times \mathbb{R}_{\geq 0}^n \right\},$$

When $p = \infty$, $\mathcal{B}(T; \ell_\infty)$ corresponds to the union of $T$ hypercubes.

## C  PROOFS OF MAIN RESULTS

In this section, we provide detailed proofs of our main results, including Theorem 3.2, Theorem 4.2, Theorem 5.1 and the argument presented in Remark 5.2.

### C.1  PROOF OF THEOREM 3.2

In order to prove Theorem 3.2, we make use of the Gaussian Isoperimetric Inequality (Sudakov & Tsirelson, 1974; Borell, 1975). The proof of such inequality can be found in Ledoux (1996).

**Lemma C.1** (Gaussian Isoperimetric Inequality). Let $(\mathbb{R}^n, \mu)$ be the $n$-dimensional Gaussian space equipped with the $\ell_2$-norm distance metric. Consider an arbitrary subset $\mathcal{E} \in \mathsf{pow}(\mathbb{R}^n)$, suppose $\mathcal{H}$ is a half space that satisfies $\mu(\mathcal{H}) = \mu(\mathcal{E})$. Then for any $\epsilon \geq 0$, we have

$$\mu(\mathcal{E}_\epsilon) \geq \mu(\mathcal{H}_\epsilon) = \Phi\big(\Phi^{-1}\big(\mu(\mathcal{E})\big) + \epsilon\big),$$

where $\Phi(\cdot)$ is the cumulative distribution function of $\mathcal{N}(0, 1)$ and $\Phi^{-1}(\cdot)$ is its inverse function.

Now we are ready to prove Theorem 3.2.

*Proof of Theorem 3.2.* To begin with, we introduce the following notations. Let $\mu_-$ be the probability measure for $\mathcal{N}(-\boldsymbol{\theta}, \sigma^2 \mathbf{I}_n)$ and $\mu_+$ be the probability measure for $\mathcal{N}(\boldsymbol{\theta}, \sigma^2 \mathbf{I}_n)$, then by definition, we have $\mu = \mu_-/2 + \mu_+/2$. Consider the optimal subset $\mathcal{E}^* = \mathrm{argmin}_{\mathcal{E} \in \mathsf{pow}(\mathcal{X})}\{\mu_\epsilon(\mathcal{E}) : \mu(\mathcal{E}) \geq \alpha\}$.

Note that the standard concentration function $h(\mu, \alpha, \epsilon)$ is monotonically increasing with respect to $\alpha$, thus $\mu(\mathcal{E}^*) = \alpha$ holds for any continuous $\mu$. Let $\alpha_- = \mu_-(\mathcal{E}^*)$ and $\alpha_+ = \mu_+(\mathcal{E}^*)$. According to the Gaussian Isoperimetric Inequality Lemma C.1, it holds for any $\epsilon \geq 0$ that

$$\mu(\mathcal{E}_\epsilon^*) = \frac{1}{2}\mu_-(\mathcal{E}_\epsilon^*) + \frac{1}{2}\mu_+(\mathcal{E}_\epsilon^*) \geq \frac{1}{2}\Phi(\Phi^{-1}(\alpha_-) + \epsilon) + \frac{1}{2}\Phi(\Phi^{-1}(\alpha_+) + \epsilon). \tag{C.1}$$

Note that the equality of (C.1) can be achieved if and only if $\mathcal{E}^*$ is a half space.

Next, we show that there always exists a half space $\mathcal{H} \in \mathsf{pow}(\mathcal{X})$ such that $\mu_-(\mathcal{H}) = \alpha_-$ and $\mu_+(\mathcal{H}) = \alpha_+$. Let $f_-(\cdot), f_+(\cdot)$ be the PDFs of $\mu_-$ and $\mu_+$ respectively. For any $\boldsymbol{x} \in \mathcal{X}$, $f_-(\boldsymbol{x})$ and $f_+(\boldsymbol{x})$ are always positive, thus we have

$$\frac{f_+(\boldsymbol{x})}{f_-(\boldsymbol{x})} = \frac{\exp\big\{ -\frac{1}{2\sigma^2}(\boldsymbol{x} - \boldsymbol{\theta})^\top(\boldsymbol{x} - \boldsymbol{\theta})\big\}}{\exp\big\{ -\frac{1}{2\sigma^2}(\boldsymbol{x} + \boldsymbol{\theta})^\top(\boldsymbol{x} + \boldsymbol{\theta})\big\}} = \exp\bigg(\frac{2\boldsymbol{\theta}^\top \boldsymbol{x}}{\sigma^2}\bigg).$$

This implies that the ratio of $f_+(\boldsymbol{x})/f_-(\boldsymbol{x})$ is monotonically increasing with respect to $\boldsymbol{\theta}^\top \boldsymbol{x}$.

Consider the following extreme half space $\mathcal{H}_- = \{\boldsymbol{x} \in \mathcal{X} : \boldsymbol{\theta}^\top \boldsymbol{x} + b \cdot \|\boldsymbol{\theta}\|_2 \leq 0\}$ such that $\mu(\mathcal{H}_-) = \alpha$. We are going to prove $\mu_-(\mathcal{H}_-) \geq \mu_-(\mathcal{E}^*) = \alpha_-$ and $\mu_+(\mathcal{H}_-) \leq \mu_+(\mathcal{E}^*) = \alpha_+$. Consider the sets $\mathcal{E}^* \cap (\mathcal{H}_-)^{\complement}$ and $(\mathcal{E}^*)^{\complement} \cap \mathcal{H}_-$, we have

$$\frac{\mu_+\big(\mathcal{E}^* \cap (\mathcal{H}_-)^{\complement}\big)}{\mu_-\big(\mathcal{E}^* \cap (\mathcal{H}_-)^{\complement}\big)} \geq \inf_{\boldsymbol{x} \in \mathcal{E}^* \cap (\mathcal{H}_-)^{\complement}} \exp\bigg(\frac{2\boldsymbol{\theta}^\top \boldsymbol{x}}{\sigma^2}\bigg) \geq \sup_{\boldsymbol{x} \in (\mathcal{E}^*)^{\complement} \cap \mathcal{H}_-} \bigg(\frac{2\boldsymbol{\theta}^\top \boldsymbol{x}}{\sigma^2}\bigg) \geq \frac{\mu_+\big((\mathcal{E}^*)^{\complement} \cap \mathcal{H}_-\big)}{\mu_-\big((\mathcal{E}^*)^{\complement} \cap \mathcal{H}_-\big)}.$$

$$\tag{C.2}$$

Note that we also have

$$\mu_+\big(\mathcal{E}^* \cap (\mathcal{H}_-)^\complement\big) + \mu_-\big(\mathcal{E}^* \cap (\mathcal{H}_-)^\complement = \mu_+\big((\mathcal{E}^*)^\complement \cap \mathcal{H}_-\big) + \mu_-\big((\mathcal{E}^*)^\complement \cap \mathcal{H}_-\big). \tag{C.3}$$

Thus, combining (C.2) and (C.3), we have

$$\mu_+\big(\mathcal{E}^* \cap (\mathcal{H}_-)^\complement\big) \geq \mu_+\big((\mathcal{E}^*)^\complement \cap \mathcal{H}_-\big) \quad \text{and} \quad \mu_-\big(\mathcal{E}^* \cap (\mathcal{H}_-)^\complement\big) \leq \mu_-\big((\mathcal{E}^*)^\complement \cap \mathcal{H}_-\big),$$

Adding the term $\mu_+\big(\mathcal{E}^* \cap \mathcal{H}_-\big)$ or $\mu_-\big(\mathcal{E}^* \cap \mathcal{H}_-\big)$ on both sides, we further have

$$\mu_+(\mathcal{H}_-) \leq \mu_+(\mathcal{E}^*) = \alpha_+ \quad \text{and} \quad \mu_-(\mathcal{H}_-) \geq \mu_-(\mathcal{E}^*) = \alpha_-.$$

On the other hand, consider the half space $\mathcal{H}_+ = \{x \in \mathcal{X} : \theta^\top x - b \cdot \|\theta\|_2 \geq 0\}$ such that $\mu(\mathcal{H}_+) = \alpha$. Based on a similar technique, we can prove

$$\mu_-(\mathcal{H}_+) \geq \mu_+(\mathcal{E}^*) = \alpha_+ \quad \text{and} \quad \mu_-(\mathcal{H}_+) \leq \mu_-(\mathcal{E}^*) = \alpha_-.$$

In addition, let $\mathcal{H} = \{x \in \mathcal{X} : w^\top x + b \leq 0\}$ be any half space such that $\mu(\mathcal{H}) = \alpha$. Since both $\mu_+$ and $\mu_-$ are continuous, as we rotate the half space (i.e., gradually increase the value of $w^\top \theta$), $\mu_-(\mathcal{H})$ and $\mu_+(\mathcal{H})$ will also change continuously. Therefore, it is guaranteed that there exists a half space $\mathcal{H} \in \text{pow}(\mathcal{X})$ such that $\mu_-(\mathcal{H}) = \alpha_-$ and $\mu_+(\mathcal{H}) = \alpha_+$. This further implies that the lower bound of (C.1) can be always be achieved.

Finally, since we have proved the optimal subset has to be a half space, the remaining task is to solve the following optimization problem:

$$\min_{\mathcal{H} \in \text{pow}(\mathcal{X})} \frac{1}{2}\Phi\big(\Phi^{-1}\big(\mu_-(\mathcal{H})\big) + \epsilon\big) + \frac{1}{2}\Phi\big(\Phi^{-1}\big(\mu_+(\mathcal{H})\big) + \epsilon\big) \tag{C.4}$$
$$\text{s.t. } \mathcal{H} = \{x \in \mathcal{X} : w^\top x + b \leq 0\} \quad \text{and} \quad \mu(\mathcal{H}) = \alpha.$$

Construct function $g(u) = \Phi\big(\Phi^{-1}(u) + \epsilon\big) + \Phi\big(\Phi^{-1}(2\alpha - u) + \epsilon\big)$, where $u \in [0, 2\alpha]$. Based on the derivative of inverse function formula, we compute the derivative of $g$ with respect to $u$ as follows

$$\begin{aligned}
\frac{\mathrm{d}g(u)}{\mathrm{d}u} &= \frac{1}{\sqrt{2\pi}}\exp\left\{-\frac{(\Phi^{-1}(u) + \epsilon)^2}{2}\right\} \cdot \frac{\mathrm{d}\Phi^{-1}(u)}{\mathrm{d}u} \\
&\quad + \frac{1}{\sqrt{2\pi}}\exp\left\{-\frac{(\Phi^{-1}(2\alpha - u) + \epsilon)^2}{2}\right\} \cdot \frac{\mathrm{d}\Phi^{-1}(2\alpha - u)}{\mathrm{d}u} \\
&= \exp\left\{-\frac{(\Phi^{-1}(u) + \epsilon)^2}{2}\right\} \cdot \exp\left\{\frac{(\Phi^{-1}(u))^2}{2}\right\} \\
&\quad - \exp\left\{-\frac{(\Phi^{-1}(2\alpha - u) + \epsilon)^2}{2}\right\} \cdot \exp\left\{\frac{(\Phi^{-1}(2\alpha - u))^2}{2}\right\} \\
&= \exp(-\epsilon^2/2) \cdot \left[\exp\big(-\epsilon\Phi^{-1}(u)\big) - \exp\big(-\epsilon\Phi^{-1}(2\alpha - u)\big)\right].
\end{aligned}$$

Noticing the term $\exp(-\epsilon\Phi^{-1}(u))$ is monotonically decreasing with respect to $u$, we then know that $g(u)$ is monotonically increasing in $[0, \alpha]$ and monotonically decreasing in $[\alpha, 2\alpha]$. Therefore, this suggests that the optimal solution to (C.4) is achieved when $\mu_-(\mathcal{H})$ reaches its maximum or its minimum. According to the previous argument regarding the range of $\alpha_-$ and $\alpha_+$, we can immediately prove the optimality results of Theorem 3.2. $\qquad\square$

## C.2 Proof of Theorem 4.2

In this section, we prove Theorem 4.2 based on techniques used in Mahloujifar et al. (2019a) for proving the connection between the standard concentration function and intrinsic robustness with respect to the set of imperfect classifiers.

*Proof of Theorem 4.2.* Let $\mathcal{E}^*$ be the optimal solution to (4.1), then $\mu(\mathcal{E}_\epsilon^*)$ corresponds to the optimal value of (4.1). We are going to show $1 - \overline{\text{AdvRob}}_\epsilon(\mathcal{F}_{\alpha,\gamma}, c) = \mu(\mathcal{E}_\epsilon^*)$ by proving both directions.

First, we prove $1 - \overline{\text{AdvRob}}_\epsilon(\mathcal{F}_{\alpha,\gamma}, c) \geq \mu(\mathcal{E}_\epsilon^*)$. Let $f$ be any classifier within $\mathcal{F}_{\alpha,\gamma}$, and $\mathcal{E}(f)$ be the corresponding error region of $f$. According to the definitions of risk and adversarial risk, we have

$$\text{Risk}(f, c) = \mu(\mathcal{E}(f)) \quad \text{and} \quad \text{AdvRisk}_\epsilon(f, c) = \mu(\mathcal{E}_\epsilon(f)),$$

where $\mathcal{E}_\epsilon(f)$ represents the $\epsilon$-expansion of $\mathcal{E}(f)$. Since $f \in \mathcal{F}_{\alpha,\gamma}$, we have

$$\text{Risk}(f,c) = \mu(\mathcal{E}(f)) \geq \alpha \ \text{ and } \ \text{LU}(\mathcal{E}(f);\mu,c,\eta) \geq \gamma.$$

Thus, by (4.1), we obtain that

$$1 - \text{AdvRob}_\epsilon(f,c) = \text{AdvRisk}_\epsilon(f,c) = \mu(\mathcal{E}_\epsilon(f)) \geq \mu(\mathcal{E}_\epsilon^*).$$

By taking the infimum over $f$ over $\mathcal{F}_{\alpha,\gamma}$ on both sides, we prove $1 - \overline{\text{AdvRob}}_\epsilon(\mathcal{F}_{\alpha,\gamma},c) \geq \mu(\mathcal{E}_\epsilon^*)$.

Next, we show that $1 - \overline{\text{AdvRob}}_\epsilon(\mathcal{F}_{\alpha,\gamma},c) \leq \mu(\mathcal{E}_\epsilon^*)$. We construct a classifier $f^*$ such that

$$f^*(\boldsymbol{x}) = c(\boldsymbol{x}) \ \text{ if } \ \boldsymbol{x} \notin \mathcal{E}^*; \ f^*(\boldsymbol{x}) \neq c(\boldsymbol{x}) \ \text{ otherwise.}$$

Note that by construction, $\mathcal{E}^*$ corresponds to the error region of $f^*$. Thus according to the definitions of risk and adversarial risk, we know

$$\text{Risk}(f^*,c) = \mu(\mathcal{E}^*) \geq \alpha \ \text{ and } \ \text{AdvRisk}_\epsilon(f^*,c) = \mu(\mathcal{E}_\epsilon^*).$$

Since $\text{LU}(\mathcal{E}^*;\mu,c,\eta) \geq \gamma$, we know the error region label uncertainty of $f^*$ is at least $\gamma$. Thus, by definition of intrinsic robustness, we know $1 - \overline{\text{AdvRob}}_\epsilon(\mathcal{F}_{\alpha,\gamma},c) \leq \text{AdvRisk}_\epsilon(f^*,c) = \mu(\mathcal{E}_\epsilon^*)$.

Finally, putting pieces together, we complete the proof. $\qquad\square$

### C.3 Proof of Theorem 5.1

*Proof of Theorem 5.1.* For simplicity, denote by $\text{lu}(\boldsymbol{x};c,\eta) = 1 - \big[\eta(\boldsymbol{x})\big]_{c(\boldsymbol{x})} + \max_{y' \neq c(\boldsymbol{x})} \big[\eta(\boldsymbol{x})\big]_{y'}$ the label uncertainty of a given input $\boldsymbol{x}$ with respect to $c(\cdot)$ and $\eta(\cdot)$. Let $\mathcal{E}$ be a subset in $\mathcal{G}$ such that $\mu(\mathcal{E}) \geq \alpha$ and $|\mu(\mathcal{E}) - \widehat{\mu}(\mathcal{E})| \leq \delta$, where $\delta$ is a constant much smaller than $\alpha$. Then according to Definition 4.1, we can decompose the estimation error of label uncertainty as:

$$\text{LU}(\mathcal{E};\mu,c,\eta) - \text{LU}(\mathcal{E};\widehat{\mu}_\mathcal{S},c,\widehat{\eta}) = \frac{1}{\mu(\mathcal{E})} \int_\mathcal{E} \text{lu}(\boldsymbol{x};c,\eta)\,d\mu - \frac{1}{\widehat{\mu}_\mathcal{S}(\mathcal{E})} \int_\mathcal{E} \text{lu}(\boldsymbol{x};c,\widehat{\eta})\,d\widehat{\mu}_\mathcal{S}$$

$$= \underbrace{\left(\frac{1}{\mu(\mathcal{E})} - \frac{1}{\widehat{\mu}_\mathcal{S}(\mathcal{E})}\right) \cdot \int_\mathcal{E} \text{lu}(\boldsymbol{x};c,\eta)\,d\mu}_{I_1}$$

$$+ \underbrace{\frac{1}{\widehat{\mu}_\mathcal{S}(\mathcal{E})} \int_\mathcal{E} \big[\text{lu}(\boldsymbol{x};c,\eta) - \text{lu}(\boldsymbol{x};c,\widehat{\eta})\big]\,d\mu}_{I_2}$$

$$+ \underbrace{\frac{1}{\widehat{\mu}_\mathcal{S}(\mathcal{E})} \left(\int_\mathcal{E} \text{lu}(\boldsymbol{x};c,\widehat{\eta})\,d\mu - \int_\mathcal{E} \text{lu}(\boldsymbol{x};c,\widehat{\eta})\,d\widehat{\mu}_\mathcal{S}\right)}_{I_3}.$$

Next, we upper bound the absolute value of the three components, respectively.

Consider the first term $I_1$. Note that $0 \leq \text{lu}(\boldsymbol{x};c,\eta) \leq 2$ for any $\boldsymbol{x} \in \mathcal{X}$, thus we have $|\int_\mathcal{E} \text{lu}(\boldsymbol{x};c,\eta)\,d\mu| \leq 2\mu(\mathcal{E})$. Therefore, we have

$$|I_1| \leq \left|\frac{1}{\mu(\mathcal{E})} - \frac{1}{\widehat{\mu}_\mathcal{S}(\mathcal{E})}\right| \cdot 2\mu(\mathcal{E}) \leq \frac{2}{\widehat{\mu}_\mathcal{S}(\mathcal{E})} \cdot |\mu(\mathcal{E}) - \widehat{\mu}_\mathcal{S}(\mathcal{E})|.$$

As for the second term $I_2$, the following inequality holds for any $\boldsymbol{x} \in \mathcal{X}$

$$|\text{lu}(\boldsymbol{x};c,\eta) - \text{lu}(\boldsymbol{x};c,\widehat{\eta})| \leq \left|\big[\eta(\boldsymbol{x}) - \widehat{\eta}(\boldsymbol{x})\big]_{c(\boldsymbol{x})}\right| + \left|\max_{y' \neq c(\boldsymbol{x})} \big[\eta(\boldsymbol{x})\big]_{y'} - \max_{y' \neq c(\boldsymbol{x})} \big[\widehat{\eta}(\boldsymbol{x})\big]_{y'}\right|$$

$$\leq \big\|\eta(\boldsymbol{x}) - \widehat{\eta}(\boldsymbol{x})\big\|_1,$$

where the second inequality holds because $|\max_i a_i - \max_i b_i| \leq \max_i |a_i - b_i|$ for any $\mathbf{a},\boldsymbol{b} \in \mathbb{R}^n$. Therefore, we can upper bound $|I_2|$ by

$$|I_2| \leq \frac{1}{\widehat{\mu}_\mathcal{S}(\mathcal{E})} \int_\mathcal{E} \big\|\eta(\boldsymbol{x}) - \widehat{\eta}(\boldsymbol{x})\big\|_1\,d\mu \leq \frac{1}{\widehat{\mu}_\mathcal{S}(\mathcal{E})} \int_\mathcal{X} \big\|\eta(\boldsymbol{x}) - \widehat{\eta}(\boldsymbol{x})\big\|_1\,d\mu \leq \frac{\delta_\eta}{\widehat{\mu}_\mathcal{S}(\mathcal{E})}.$$

For the last term $I_3$, since $0 \leq \mathrm{lu}(\boldsymbol{x}; c, \eta) \leq 2$ holds for any $\boldsymbol{x} \in \mathcal{X}$, we have

$$|I_3| \leq \frac{2}{\widehat{\mu}_{\mathcal{S}}(\mathcal{E})} \cdot \big|\mu(\mathcal{E}) - \widehat{\mu}_{\mathcal{S}}(\mathcal{E})\big|.$$

Finally, putting pieces together, we have

$$|\mathrm{LU}(\mathcal{E}; \mu, c, \eta) - \mathrm{LU}(\mathcal{E}; \widehat{\mu}_{\mathcal{S}}, c, \widehat{\eta})| \leq \frac{4}{\widehat{\mu}_{\mathcal{S}}(\mathcal{E})} \cdot \big|\mu(\mathcal{E}) - \widehat{\mu}_{\mathcal{S}}(\mathcal{E})\big| + \frac{\delta_\eta}{\widehat{\mu}_{\mathcal{S}}(\mathcal{E})} \leq \frac{4\delta + \delta_\eta}{\alpha - \delta},$$

provided $\mu(\mathcal{E}) \geq \alpha$ and $|\mu(\mathcal{E}) - \widehat{\mu}_{\mathcal{S}}(\mathcal{E})| \leq \delta$. Making use of the definition of complexity penalty for $\mathcal{G}$ completes the proof of Theorem 5.1. $\qquad\square$

## C.4 PROOF OF REMARK 5.2

Before presenting the proofs, we first lay out the formal statement of Remark 5.2 in Theorem C.2. The proof technique of Theorem C.2 is inspired by Theorem 3.5 in Mahloujifar et al. (2019b).

**Theorem C.2** (Formal Statement of Remark 5.2). Consider the input metric probability space $(\mathcal{X}, \mu, \Delta)$, the concept function $c$ and the label distribution function $\eta$. Let $\{\mathcal{G}(T)\}_{T \in \mathbb{N}}$ be a series of collection of subsets over $\mathcal{X}$. For any $T \in \mathbb{N}$, assume $\phi^T$ and $\phi_\epsilon^T$ are complexity penalties for $\mathcal{G}(T)$ and $\mathcal{G}_\epsilon(T)$ respectively, and $\widehat{\eta}$ is a function such that $\int_{\mathcal{X}} \|\widehat{\eta}(\boldsymbol{x}) - \eta(\boldsymbol{x})\|_1 d\boldsymbol{x} \leq \delta_\eta$.

Define $h(\mu, c, \eta, \alpha, \gamma, \epsilon, \mathcal{G}) = \inf_{\mathcal{E} \in \mathcal{G}} \{\mu(\mathcal{E}_\epsilon) : \mu(\mathcal{E}) \geq \alpha, \mathrm{LU}(\mathcal{E}; \mu, c, \eta) \geq \gamma\}$ to be the constrained concentration function. We simply write $h(\mu, c, \eta, \alpha, \gamma, \epsilon)$ when $\mathcal{G} = \mathrm{pow}(\mathcal{X})$. Given a sequence of datasets $\{S_T\}_{T \in \mathbb{N}}$, where $S_T$ consists of $m(T)$ i.i.d. samples from $\mu$ and a sequence of real numbers $\{\delta(T)\}_{T \in \mathbb{N}}$ with $\delta(T) \in (0, \alpha/2)$, if the following assumptions holds:

1. $\sum_{T=1}^{\infty} \phi^T(m(T), \delta(T)) < \infty$

2. $\sum_{T=1}^{\infty} \phi_\epsilon^T(m(T), \delta(T)) < \infty$

3. $\lim_{T \to \infty} \delta(T) = 0$

4. $\lim_{T \to \infty} h(\mu, c, \eta, \alpha, \gamma, \epsilon, \mathcal{G}(T)) = h(\mu, c, \eta, \alpha, \gamma, \epsilon)$ [1]

5. $h$ is locally continuous w.r.t. $\alpha$ and $\gamma$ at $(\mu, c, \eta, \alpha, \gamma \pm \delta_\eta/\alpha, \epsilon, \mathrm{pow}(\mathcal{X}))$,

then with probability 1, we have

$$h(\mu, c, \eta, \alpha, \gamma - \delta_\eta/\alpha, \epsilon) \leq \lim_{T \to \infty} h(\mu_{S_T}, c, \widehat{\eta}, \alpha, \gamma, \epsilon, \mathcal{G}(T)) \leq h(\mu, c, \eta, \alpha, \gamma + \delta_\eta/\alpha, \epsilon).$$

To prove Theorem C.2, we use the following theorem regarding the generalization of concentration under label uncertainty constraints. The proof of Theorem C.3 is provided in Appendix C.5.

**Theorem C.3** (Generalization of Concentration). Let $(\mathcal{X}, \mu, \Delta)$ be a metric probability space and $\mathcal{G} \subseteq \mathrm{pow}(\mathcal{X})$. Define $h(\mu, c, \eta, \alpha, \gamma, \epsilon, \mathcal{G}) = \inf_{\mathcal{E} \in \mathcal{G}} \{\mu(\mathcal{E}_\epsilon) : \mu(\mathcal{E}) \geq \alpha, \mathrm{LU}(\mathcal{E}; \mu, c, \eta) \geq \gamma\}$ as the generalized concentration function under label uncertainty constraints. Then, under the same setting of Theorem 5.1, for any $\gamma, \epsilon \in [0, 1]$, $\alpha \in (0, 1]$ and $\delta \in (0, \alpha/2)$, we have

$$\Pr_{\mathcal{S} \leftarrow \mu^m} \big[ h(\mu, c, \eta, \alpha - \delta, \gamma - \delta', \epsilon, \mathcal{G}) - \delta \leq h(\widehat{\mu}_{\mathcal{S}}, c, \widehat{\eta}, \alpha, \gamma, \epsilon, \mathcal{G})$$

$$\leq h(\mu, c, \eta, \alpha + \delta, \gamma + \delta', \epsilon, \mathcal{G}) + \delta \big] \geq 1 - 6\phi(m, \delta) - 2\phi_\epsilon(m, \delta),$$

where $\delta' = (4\delta + \delta_\eta)/(\alpha - 2\delta)$ and $\phi_\epsilon$ is the complexity penalty for $\mathcal{G}_\epsilon$.

In addition, we also make use of the Borel-Cantelli Lemma to prove Theorem C.2.

**Lemma C.4** (Borel-Cantelli Lemma). Let $\{E_T\}_{T \in \mathbb{N}}$ be a series of events such that $\sum_{T=1}^{\infty} \Pr[E_T] < \infty$. Then with probability 1, only finite number of events will occur.

Now we are ready to prove Theorem C.2.

---

[1] It is worth nothing that this assumption is satisfied for any family of collections of subsets that is a universal approximator, such as kernel SVMs and decision trees.

*Proof of Theorem C.2.* Let $E_T$ be the event such that

$$h\big(\mu, c, \eta, \alpha - \delta(T), \gamma - \delta'(T), \epsilon, \mathcal{G}(T)\big) - \delta(T) > h\big(\widehat{\mu}_{\mathcal{S}_T}, c, \widehat{\eta}, \alpha, \gamma, \epsilon, \mathcal{G}(T)\big) \text{ or}$$
$$h\big(\mu, c, \eta, \alpha + \delta(T), \gamma + \delta'(T), \epsilon, \mathcal{G}(T)\big) + \delta(T) < h\big(\widehat{\mu}_{\mathcal{S}_T}, c, \widehat{\eta}, \alpha, \gamma, \epsilon, \mathcal{G}(T)\big),$$

$\delta'(T) = (4\delta(T) + \delta_\eta)/(\alpha - 2\delta(T))$ for any $T \in \mathbb{N}$. Since $\delta(T) < \alpha/2$, thus according to Theorem C.3, for any $T \in \mathbb{N}$, we have

$$\Pr[E_T] \leq 6\phi^T(m(T), \delta(T)) + 2\phi_\epsilon^T(m(T), \delta(T))).$$

By Assumptions 1 and 2, this further implies

$$\sum_{T=1}^{\infty} \Pr[E_T] \leq 6 \sum_{T=1}^{\infty} \phi^T(m(T), \delta(T)) + 2 \sum_{T=1}^{\infty} \phi_\epsilon^T(m(T), \delta(T))) < \infty.$$

Thus according to Lemma C.4, we know that there exists some $j \in \mathbb{N}$ such that for all $T \geq j$,

$$h\big(\mu, c, \eta, \alpha - \delta(T), \gamma - \delta'(T), \epsilon, \mathcal{G}(T)\big) - \delta(T) \leq h(\widehat{\mu}_{\mathcal{S}_T}, c, \widehat{\eta}, \alpha, \gamma, \epsilon)$$
$$\leq h\big(\mu, c, \eta, \alpha + \delta(T), \gamma + \delta'(T), \epsilon, \mathcal{G}(T)\big) + \delta(T), \qquad (C.5)$$

holds with probability 1. In addition, by Assumptions 3, 4 and 5, we have

$$\lim_{T \to \infty} h\big(\mu, c, \eta, \alpha - \delta(T), \gamma - \delta'(T), \epsilon, \mathcal{G}(T)\big)$$
$$= \lim_{T_1 \to \infty} \lim_{T_2 \to \infty} h\big(\mu, c, \eta, \alpha - \delta(T_1), \gamma - \delta'(T_1), \epsilon, \mathcal{G}(T_2)\big)$$
$$= \lim_{T_1 \to \infty} h\big(\mu, c, \eta, \alpha - \delta(T_1), \gamma - \delta'(T_1), \epsilon\big)$$
$$= h\big(\mu, c, \eta, \alpha, \gamma - \delta_\eta/\alpha, \epsilon\big),$$

where the second equality is due to Assumption 4 and the last equality is due to Assumptions 3 and 5. Similarly, we have

$$\lim_{T \to \infty} h\big(\mu, c, \eta, \alpha + \delta(T), \gamma + \delta'(T), \epsilon, \mathcal{G}(T)\big) = h\big(\mu, c, \eta, \alpha, \gamma + \delta_\eta/\alpha, \epsilon\big).$$

Therefore, let $T$ goes to $\infty$ in (C.5), we have

$$h\big(\mu, c, \eta, \alpha, \gamma - \delta_\eta/\alpha, \epsilon\big) \leq \lim_{T \to \infty} h\big(\widehat{\mu}_{\mathcal{S}_T}, c, \widehat{\eta}, \alpha, \gamma, \epsilon, \mathcal{G}(T)\big) \leq h\big(\mu, c, \eta, \alpha, \gamma + \delta_\eta/\alpha, \epsilon\big).$$

$\square$

## C.5   PROOF OF GENERALIZATION OF CONCENTRATION THEOREM

*Proof of Theorem C.3.* First, we introduce some notation. Let $h(\mu, c, \eta, \alpha, \gamma, \epsilon, \mathcal{G})$ be the optimal value and $g(\mu, c, \eta, \alpha, \gamma, \epsilon, \mathcal{G})$ be the optimal solution with respect to the following generalized concentration of measure problem with label uncertainty constraint:

$$\underset{\mathcal{E} \in \mathcal{G}}{\text{minimize}} \ \mu(\mathcal{E}_\epsilon) \quad \text{subject to} \quad \mu(\mathcal{E}) \geq \alpha \ \text{and} \ \text{LU}(\mathcal{E}; \mu, c, \eta) \geq \gamma. \qquad (C.6)$$

Note that the difference between (C.6) and (4.1) is that the feasible set of $\mathcal{E}$ is restricted to some collection of subsets $\mathcal{G} \subseteq \text{pow}(\mathcal{X})$. Correspondingly, we let $h(\widehat{\mu}_{\mathcal{S}}, c, \widehat{\eta}, \alpha, \gamma, \epsilon, \mathcal{G})$ and $g(\widehat{\mu}_{\mathcal{S}}, c, \widehat{\eta}, \alpha, \gamma, \epsilon, \mathcal{G})$ be the optimal value and optimal solution with respect to the empirical optimization problem (5.1).

Let $\mathcal{E} = g(\mu, c, \eta, \alpha + \delta, \gamma + \delta', \epsilon, \mathcal{G})$ and $\widehat{\mathcal{E}} = g(\widehat{\mu}_{\mathcal{S}}, c, \widehat{\eta}, \alpha, \gamma, \epsilon, \mathcal{G})$, where $\delta'$ will be specified later. Note that when these optimal sets do not exist, we can select a set for which the expansion is arbitrarily close to the optimum, then every step of the proof will apply to this variant. According to the definition of complexity penalty, we have

$$\Pr_{\mathcal{S} \leftarrow \mu^m}\big[|\widehat{\mu}_{\mathcal{S}}(\widehat{\mathcal{E}}) - \mu(\widehat{\mathcal{E}})| \geq \delta\big] \leq \phi(m, \delta). \qquad (C.7)$$

Since $\widehat{\mu}_{\mathcal{S}}(\widehat{\mathcal{E}}) \geq \alpha$ by definition, (C.7) implies that

$$\Pr_{\mathcal{S} \leftarrow \mu^m}\big[\mu(\widehat{\mathcal{E}}) \leq \alpha - \delta\big] \leq \phi(m, \delta). \qquad (C.8)$$

In addition, according to Theorem 5.1, for any $\delta \in (0, \alpha/2)$, we have

$$\Pr_{\mathcal{S} \leftarrow \mu^m} \left[ \left| \mathrm{LU}(\widehat{\mathcal{E}}; \mu, c, \eta) - \mathrm{LU}(\widehat{\mathcal{E}}; \widehat{\mu}_{\mathcal{S}}, c, \widehat{\eta}) \right| \leq \frac{4\delta + \delta_\eta}{\alpha - 2\delta} \right] \leq 2\phi(m, \delta), \tag{C.9}$$

where the inequality holds because of (C.8) and the union bound. Since $\mathrm{LU}(\widehat{\mathcal{E}}; \widehat{\mu}_{\mathcal{S}}, c, \widehat{\eta}) \geq \gamma$ by definition, (C.9) implies that

$$\Pr_{\mathcal{S} \leftarrow \mu^m} \left[ \mathrm{LU}(\widehat{\mathcal{E}}; \mu, c, \eta) \leq \gamma - \frac{4\delta + \delta_\eta}{\alpha - 2\delta} \right] \leq 2\phi(m, \delta). \tag{C.10}$$

Based on the definition of the concentration function $h$, combining (C.8) and (C.10) and making use of the union bound, we have

$$\Pr_{\mathcal{S} \leftarrow \mu^m} \left[ \mu(\widehat{\mathcal{E}}_\epsilon) \leq h(\mu, c, \eta, \alpha - \delta, \gamma - \delta', \epsilon, \mathcal{G}) \right] \leq 3\phi(m, \delta), \tag{C.11}$$

where we set $\delta' = \frac{4\delta + \delta_\eta}{\alpha - 2\delta}$. Note that according to the definition of $\phi_\epsilon$, we have

$$\Pr_{\mathcal{S} \leftarrow \mu^m} \left[ |\mu(\widehat{\mathcal{E}}_\epsilon) - \mu_{\mathcal{S}}(\widehat{\mathcal{E}}_\epsilon)| \leq \delta \right] \leq \phi_\epsilon(m, \delta), \tag{C.12}$$

thus combining (C.11) and (C.12) by union bound, we have

$$\Pr_{\mathcal{S} \leftarrow \mu^m} \left[ \widehat{\mu}_{\mathcal{S}}(\widehat{\mathcal{E}}_\epsilon) \leq h(\mu, c, \eta, \alpha - \delta, \gamma - \delta', \epsilon, \mathcal{G}) - \delta \right] \leq 3\phi(m, \delta) + \phi_\epsilon(m, \delta). \tag{C.13}$$

This completes the proof of one-sided inequality of Theorem C.3. The other side of Theorem C.3 can be proved using the same technique. In particular, we have

$$\Pr_{\mathcal{S} \leftarrow \mu^m} \left[ \widehat{\mu}_{\mathcal{S}}(\widehat{\mathcal{E}}_\epsilon) \geq h(\mu, c, \eta, \alpha + \delta, \gamma + \delta', \epsilon, \mathcal{G}) + \delta \right] \leq 3\phi(m, \delta) + \phi_\epsilon(m, \delta). \tag{C.14}$$

Combining (C.13) and (C.14) by union bound completes the proof. $\qquad\square$

## D  HEURISTIC SEARCH ALGORITHM

The pseudocode of the heuristic search algorithm for the empirical label uncertainty constrained concentration problem (5.1) is shown in Algorithm 1.

---

**Algorithm 1:** Heuristic Search for Robust Error Region under $\ell_p (p \in \{2, \infty\})$

---
**Input** : a set of labeled inputs $\{\boldsymbol{x}, c(\boldsymbol{x}), \widehat{\eta}(\boldsymbol{x})\}_{\boldsymbol{x} \in \mathcal{S}}$, parameters $\alpha, \gamma, \epsilon, T$

1   $\widehat{\mathcal{E}} \leftarrow \{\}, \quad \widehat{\mathcal{S}}_{\mathrm{init}} \leftarrow \{\}, \quad \widehat{\mathcal{S}}_{\mathrm{exp}} \leftarrow \{\};$

2   **for** $t = 1, 2, \ldots, T$ **do**

3      $k_{\mathrm{lower}} \leftarrow \lceil (\alpha|\mathcal{S}| - |\widehat{\mathcal{S}}_{\mathrm{init}}|)/(T - t + 1) \rceil, \quad k_{\mathrm{upper}} \leftarrow (\alpha|\mathcal{S}| - |\widehat{\mathcal{S}}_{\mathrm{init}}|);$

4      $\Omega \leftarrow \{\};$

5      **for** $\boldsymbol{u} \in \mathcal{S}$ **do**

6         **for** $k \in [k_{lower}, k_{upper}]$ **do**

7            $r_k(\boldsymbol{u}) \leftarrow$ compute the $\ell_p$ distance from $\boldsymbol{u}$ to the $k$-th nearest neighbour in $\mathcal{S} \setminus \widehat{\mathcal{S}}_{\mathrm{init}};$

8            $\mathcal{S}_{\mathrm{init}}(\boldsymbol{u}, k) \leftarrow \{\boldsymbol{x} \in \mathcal{S} \setminus \widehat{\mathcal{S}}_{\mathrm{init}} : \|\boldsymbol{x} - \boldsymbol{u}\|_2 \leq r_k(\boldsymbol{u})\};$

9            $\mathcal{S}_{\mathrm{exp}}(\boldsymbol{u}, k) \leftarrow \{\boldsymbol{x} \in \mathcal{S} \setminus \widehat{\mathcal{S}}_{\mathrm{exp}} : \|\boldsymbol{x} - \boldsymbol{u}\|_2 \leq r_k(\boldsymbol{u}) + \epsilon\};$

10           **if** $\mathrm{LU}(\mathcal{S}_{\mathrm{init}}(\boldsymbol{u}, k), \widehat{\mu}_{\mathcal{S}}, c, \widehat{\eta}) \geq \gamma$ **then**

11             insert $(\boldsymbol{u}, k)$ into $\Omega$

12      $(\widehat{\boldsymbol{u}}, \widehat{k}) \leftarrow \operatorname{argmin}_{(\boldsymbol{u}, k) \in \Omega} \{|\mathcal{S}_{\mathrm{exp}}(\boldsymbol{u}, k)| - |\mathcal{S}_{\mathrm{init}}(\boldsymbol{u}, k)|\};$

13      $\widehat{\mathcal{E}} \leftarrow \widehat{\mathcal{E}} \cup \mathrm{Ball}(\widehat{\boldsymbol{u}}, r_{\widehat{k}}(\widehat{\boldsymbol{u}}));$

14      $\widehat{\mathcal{S}}_{\mathrm{init}} \leftarrow \widehat{\mathcal{S}}_{\mathrm{init}} \cup \mathcal{S}_{\mathrm{init}}(\widehat{\boldsymbol{u}}, \widehat{k}), \quad \widehat{\mathcal{S}}_{\mathrm{exp}} \leftarrow \widehat{\mathcal{S}}_{\mathrm{exp}} \cup \mathcal{S}_{\mathrm{exp}}(\widehat{\boldsymbol{u}}, \widehat{k});$

**Output** : $\widehat{\mathcal{E}}$

---

# E DETAILED EXPERIMENTAL SETTINGS

In this section, we specify the details of the experiments presented in Section 6. The robustness results of all the adversarially-trained models from RobustBench (Croce et al., 2020) are evaluated using the auto attack (Croce & Hein, 2020). All of our experiments are conducted using a GPU server with a NVIDIA GeForce RTX 2080 Ti Graphics card.

**Error Region Label Uncertainty.** We explain the experimental details of Figure 3. For standard trained classifiers, we implemented five neural network architecture, including a 4-layer neural net with two convolutional layers and two fully-connected layers (*small*), a 7-layer neural net with four convolutional layers and three fully-connected layers (*large*), a ResNet-18 architecture (*resnet18*), ResNet-50 architecture (*resnet50*) and a WideResNet-34-10 architecture (*wideresnet*). We trained the *small* and *large* model using a Adam optimizer with initial learning rate $0.005$, whereas we trained the *resnet18*, *resnet50* and *wideresnet* model using a SGD optimizer with initial learning rate $0.01$. All models are trained using a piece-wise learning rate schedule with a decaying factor of $10$ at epoch $50$ and epoch $75$, respectively. For Figure 3(a), we plotted the label uncertainty and standard risk for the intermediate models obtained at epochs $5, 10, \ldots, 100$ for each architecture. In addition, we also randomly selected different subsets of inputs with empirical measure of $0.05, 0.10, \ldots 0.95$ and plotted their corresponding label uncertainty with error bars.

For adversarially trained classifiers, we implemented the vanilla adversarial training method (Mądry et al., 2018) and the adversarial training method with adversarial weight perturbation (Wu et al., 2020), which are denoted as *AT* and *AT-AWP* in Figure 3(b) respectively. Both ResNet-18 (*resnet18*) and WideResNet-34-10 (*wideresnet*) architecture are implemented for each training method. A 10-step PGD attack (PGD-10) with step size $2/255$ and maximum perturbation size $8/255$ is used for each model during training. In addition, each model is trained for 200 epochs using a SGD optimizer with initial learning rate $0.1$ and piece-wise learning rate schedule with a decaying factor of $10$ at epoch $100$ and epoch $150$. We record the intermediate models at epoch $10, 20, \ldots, 200$ respectively.

**Estimation of Intrinsic Robustness.** For Figure 4, we first conduct a $50/50$ train-test split over the $10,000$ CIFAR-10 test images, then run Algorithm 1 for each setting on the training dataset to obtain the optimal subset. Here, we choose the value of $\alpha \in \{0.01, 0.02, \ldots, 0.15\}$ and tune the parameter $T$ for each $\alpha$ parameter. Next, we evaluate the empirical measure of the optimally-searched subset (denoted by *empirical risk* in Figure 4) and the empirical measure of its $\epsilon$-expansion using the testing dataset, and translate it into an intrinsic robustness estimate. Finally, we plot the empirical risk and the estimated intrinsic robustness for each parameter setting in Figure 4.

# F ADDITIONAL EXPERIMENTS

This appendix provides additional experimental results, supporting our arguments in Section 6.

**Visualization of label uncertainty.** Figure 6 shows some CIFAR-10 images with the original CIFAR-10 labels and the CIFAR-10H human uncertainty labels. The label uncertainty score is computed based on Definition 4.1 and provided under each image.

There are a few examples with high label uncertainty, whose CIFAR-10 label contradicts with the CIFAR-10H soft label (see the first two images in Figure 6(a)), indicating they are actually mislabeled. The images with uncertainty scores around $1.0$ do appear to be images that are difficult for human to recognize, whereas images with lowest uncertainty scores look clearly representative of the labeled class. These observations show the usefulness of the proposed label uncertainty definition.

**Estimation of Intrinsic Robustness.** Table 1 summarizes our estimated intrinsic robustness limits produced by Algorithm 1 for different hyperparameter settings. In particular, we set $\alpha = 0.05$ and $\gamma = 0.17$ to roughly reflect the standard error and the label uncertainty of the error regions with respect to the state-of-the-art classification models (see Figure 3), use $\epsilon \in \{4/255, 8/255, 16/255\}$ for $\ell_\infty$ and $\epsilon \in \{0.5, 1.0, 1.5\}$ for $\ell_2$. Note that we also compare our estimate with the intrinsic robustness limit implied by the standard concentration by setting $\gamma = 0$ for each setting.

We perform a $50/50$ train-test split on the CIFAR-10 test images: we obtain the optimal subset with the smallest $\epsilon$-expansion on the training dataset based on Algorithm 1 and evaluate it on the testing

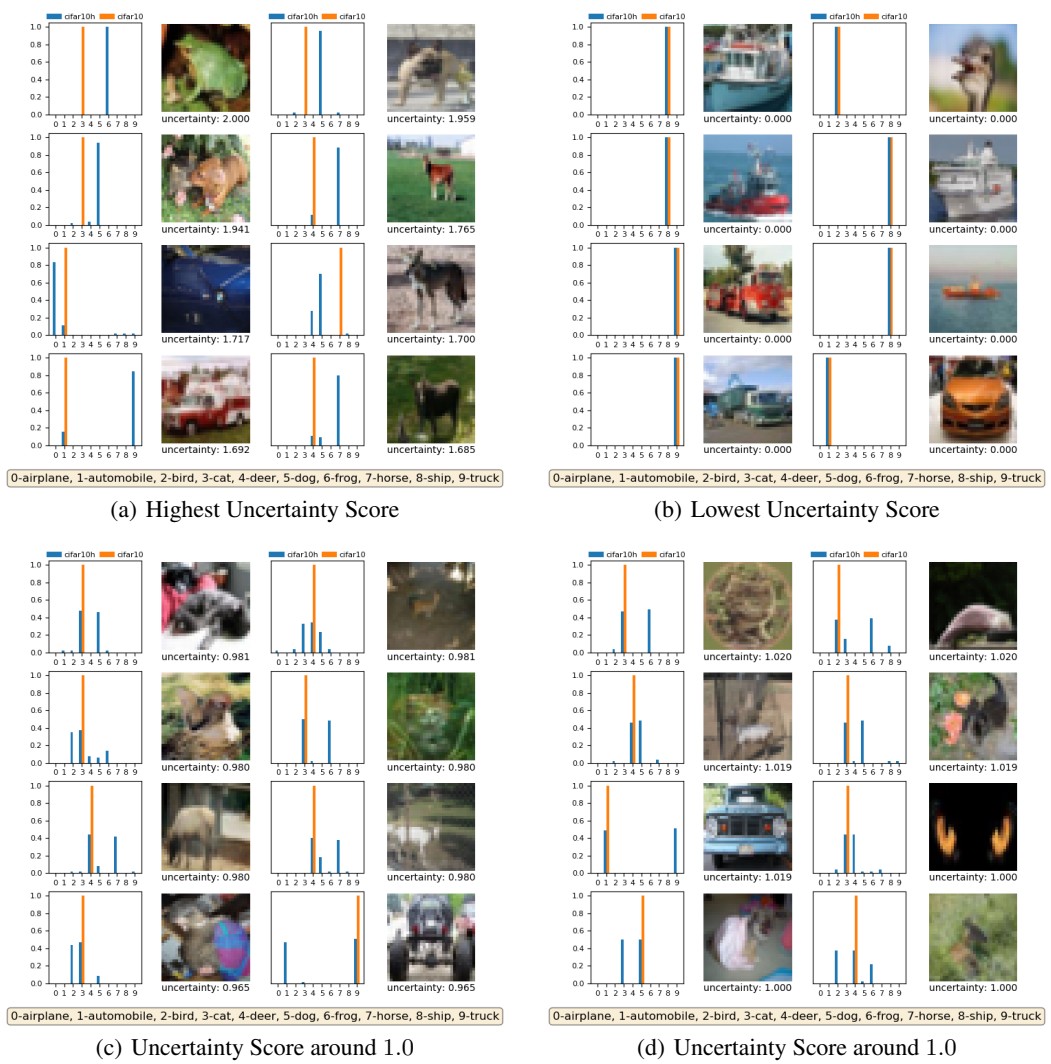

Figure 6: Illustration of human uncertainty labels and label uncertainty of CIFAR-10 test images. Each subfigure shows a group of images with a certain level of uncertainty score.

dataset. We report both the empirical measure of the optimally-found subset (*Empirical Risk* in Table 1), and the translated intrinsic robustness estimate. These results show that our estimation of intrinsic robustness generalizes from the training data to the testing data, and support the argument that our estimate is a more accurate characterization of intrinsic robustness compared with standard one.

# G   ESTIMATING LABEL ERRORS USING CONFIDENT LEARNING

The proposed concentration estimation framework relies on the knowledge of human soft labels to determine which example has label uncertainty exceeding a certainty threshold. Since most machine learning datasets do not provide such information like CIFAR-10H, this raises the question of how to extend our method to the setting where human soft labels are unavailable.

We make an initial attempt to address the aforementioned issue using the confident learning approach of Northcutt et al. (2021b). Their goal was to identify label errors for a dataset, which is closely related to label uncertainty. The method first computes a confidence joint matrix based on the predicted probabilities of a pretrained classifier, then selects the top examples based on a ranking rule, such as self-confidence or max margin. If we are able to approximate human label uncertainty from the raw

Table 1: Summary of the main results using our method for different settings on CIFAR-10 dataset. We conduct 5 repeated trials for each setting to record the mean statistics and its standard deviation.

| Metric | $\alpha$ | $\epsilon$ | $\gamma$ | $T$ | Empirical Risk (%) | | Intrinsic Robustness (%) | |
|---|---|---|---|---|---|---|---|---|
| | | | | | training | testing | training | testing |
| $\ell_\infty$ | 0.05 | 4/255 | 0.0 | 5 | $5.80 \pm 0.04$ | $4.50 \pm 0.21$ | $93.48 \pm 0.10$ | $93.86 \pm 0.26$ |
| | | | 0.17 | 5 | $5.84 \pm 0.10$ | $5.06 \pm 0.82$ | $92.03 \pm 0.45$ | $92.61 \pm 1.12$ |
| | | 8/255 | 0.0 | 10 | $5.77 \pm 0.01$ | $4.76 \pm 0.27$ | $92.89 \pm 0.11$ | $92.36 \pm 0.33$ |
| | | | 0.17 | 10 | $5.77 \pm 0.02$ | $4.85 \pm 0.58$ | $90.91 \pm 0.53$ | $90.98 \pm 1.03$ |
| | | 16/255 | 0.0 | 5 | $5.68 \pm 0.04$ | $5.30 \pm 0.33$ | $88.44 \pm 0.47$ | $87.89 \pm 1.24$ |
| | | | 0.17 | 5 | $5.67 \pm 0.25$ | $4.79 \pm 0.75$ | $81.96 \pm 1.69$ | $83.83 \pm 2.37$ |
| $\ell_2$ | 0.05 | 0.5 | 0.0 | 5 | $5.76 \pm 0.00$ | $5.41 \pm 0.60$ | $93.78 \pm 0.10$ | $93.51 \pm 0.67$ |
| | | | 0.17 | 5 | $5.76 \pm 0.00$ | $5.36 \pm 0.14$ | $91.89 \pm 0.38$ | $91.70 \pm 0.49$ |
| | | 1.0 | 0.0 | 5 | $5.76 \pm 0.00$ | $6.00 \pm 0.50$ | $92.93 \pm 0.06$ | $92.22 \pm 0.55$ |
| | | | 0.17 | 5 | $5.76 \pm 0.00$ | $5.32 \pm 0.29$ | $87.86 \pm 0.79$ | $87.75 \pm 0.58$ |
| | | 1.5 | 0.0 | 5 | $5.76 \pm 0.00$ | $5.67 \pm 0.56$ | $91.98 \pm 0.13$ | $91.82 \pm 0.65$ |
| | | | 0.17 | 5 | $5.76 \pm 0.00$ | $5.69 \pm 0.45$ | $83.33 \pm 2.04$ | $82.87 \pm 2.50$ |

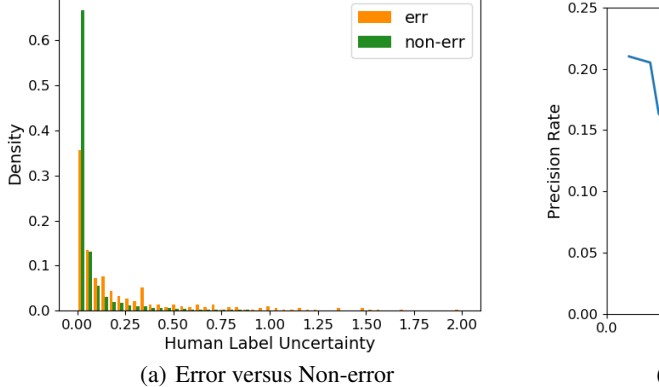

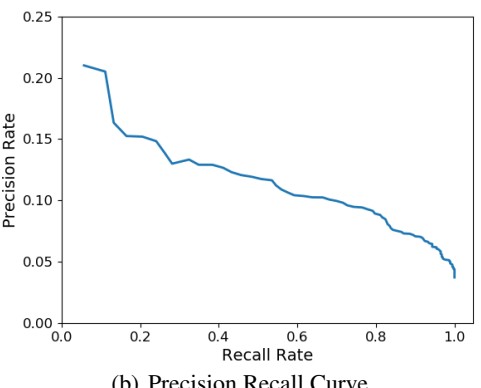

(a) Error versus Non-error  (b) Precision Recall Curve

Figure 7: Illustration of misalignment label errors recognized by human and those identified by confident learning (a) Distribution of human label uncertainty between errors and non-errors estimated using confident learning; (b) Precision-recall curve for estimating the set of examples with human label uncertainty exceeding 0.5.

inputs and labels, or identify the set of examples with high label uncertainty, then we can immediately adapt our proposed framework by leveraging such estimated results. However, we observe only a weak correlation between the set of label errors that are produced by confident learning and the set of examples with high human label uncertainty.

We conduct the experiments on CIFAR-10 and identify the set of label errors based on confident learning. We train a ResNet-50 based classification model on the CIFAR-10 training data, and select examples in the CIFAR-10 test dataset as labeling errors using the best ranking method suggested in Northcutt et al. (2021b). Figure 7(a) compares the distribution of human label uncertainty (based on the human soft labels from CIFAR-10H) between the set of estimated label error and non-errors. Although the set of examples estimated as label error have relative higher human label uncertainty compared with non-errors, there exist over 30% of estimated label errors have 0 label uncertainty for human annotators. It implies that there is a mismatch between label errors identified by human and that estimated using confident learning techniques. This is further confirmed by the precision-recall curve presented in Figure 7(b). We treat examples with human label uncertainty exceeding 0.5 as the 'ground-truth' uncertain images, and vary the size of produced set of label errors to plot the precision and recall curve. The fact that precision rate is uniformly lower than 0.25, indicating that over 75% of the estimate error examples have human label uncertainty less than 0.5.

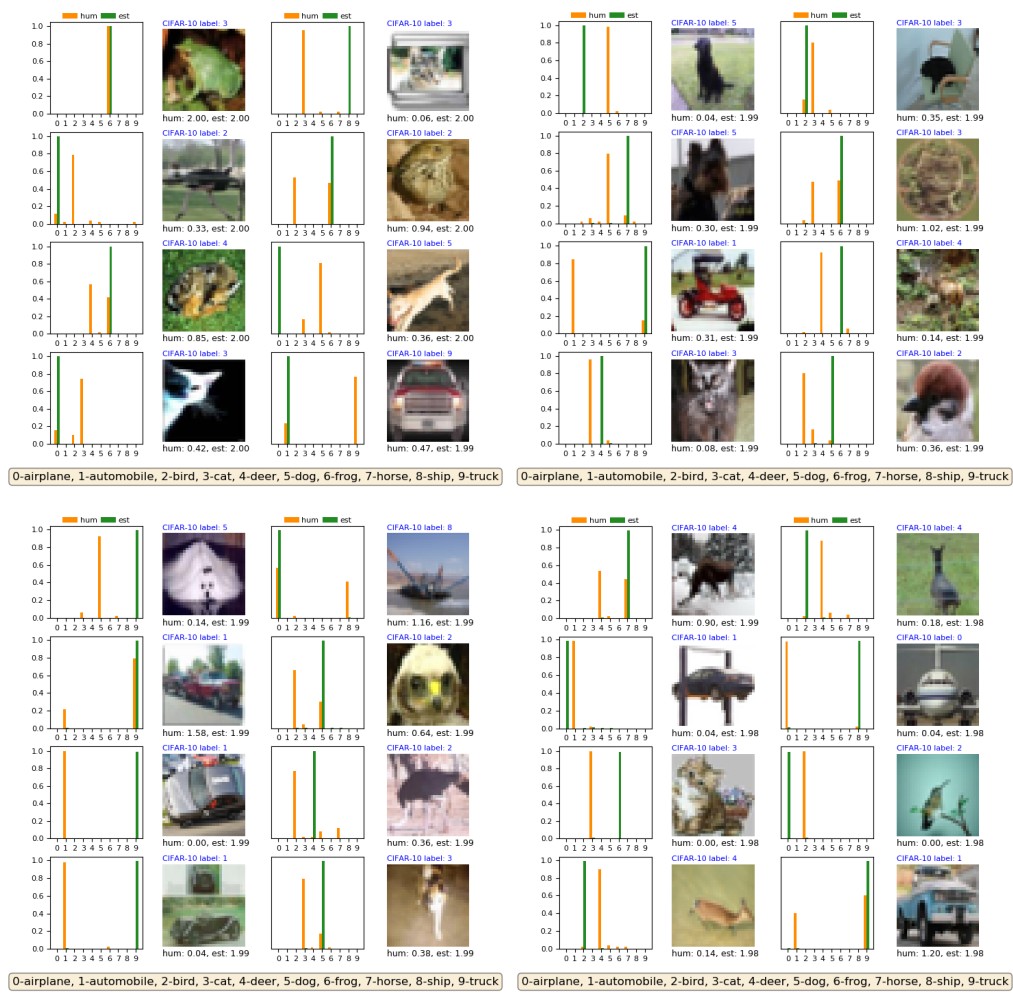

Figure 8: Visualization of label distribution of top uncertain CIFAR-10 test images estimated using a confident learning approach. Both human and estimated label distribution are plotted in each figure. The corresponding label uncertainty scores are computed and provided under each image, while the original CIFAR-10 label is highlighted in blue above each image.

Figure 8 visualizes the human label distribution and estimated label distribution on CIFAR-10. We compute the estimated label uncertainty of each CIFAR-10 testing examples by replacing the human label distribution with the predicted probabilities of the trained model. It can be seen that there exists a misalignment between the human label distribution and the distribution estimated using some neural network. This again confirms that label errors produced by confident learning are not guaranteed to be examples that are difficult for humans.

