# OpenReview forum: "Understanding Intrinsic Robustness Using Label Uncertainty"
_ICLR.cc/2022/Conference — ICLR 2022 Poster_

### Official Review · Reviewer_jR8K · 2021-10-21

**Correctness:** 3
**Technical Novelty And Significance:** 3
**Empirical Novelty And Significance:** 3
**Recommendation:** 6
**Confidence:** 3

**Main Review:**

**Strengths**
- The motivation of this paper is very clear. The authors point out the issues of the prior methods clearly. Besides, this paper explains why the class labels are important to be involved well.
- The writing and organization of this paper are great, which makes it easy to follow.
- The theoretical analysis is detailed and convincing.

**Weaknesses & Questions**
- Part of the explanation is not very detailed and confused. For example, the definition of label uncertainty in this paper is not very intuitive. It is expected for some explanations at a high level.
- Could the claims of this paper be used to improve the performance of adversarial attacks and defenses?
- I am concerned about the availability of soft labels and the effectiveness of confident learning methods for estimating label uncertainty. This may limit the application of the proposed method in the real world.

**Summary Of The Paper:**

This paper focuses on adversarial learning, which is one of the hottest topics in machine/deep learning. The authors are concerned that the standard concentration of measure problem in prior works cannot capture realistic intrinsic robustness well. In this paper, the information of class labels is therefore introduced into intrinsic robustness limits. Both theoretical analysis and experimental results are provided.

**Summary Of The Review:**

This paper focuses on an important problem in adversarial learning. The motivation, writing, and theoretical analysis are appreciated, though there are some mentioned issues needed to be addressed. Therefore, in my view, this paper reaches the acceptance line of ICLR. I recommend accepting it.

---

> ### Author Response · Authors · 2021-11-16
> **Response to Reviewer jR8K**
>
> Thank you for putting the time and effort into making our manuscript better. We are delighted that the reviewer found the contributions of our work important. Below, we provide clarifications to the questions raised by the reviewer.
>
> __Intuitive explanation of label uncertainty:__ Given an input region $\mathcal{E}$, label uncertainty is introduced to capture the average uncertainty level with respect to the label assignments of the concept function $c$ for the inputs in $\mathcal{E}$. For a given example $x$, it characterizes the degree of difference between its assigned label $c(x)$ and the underlying conditional distribution of $P(Y|X=x)$. For image classification tasks that involve human annotations, an example with a high label uncertainty implies that it is either mislabeled by $c$ or inherently ambiguous to classify for a human annotator (see Figure 1(a) for an illustration on CIFAR-10). We hypothesize that examples with high label uncertainty are expected to be more likely to be misclassified by any state-of-the-art classifiers, thus label uncertainty should be incorporated to better understand the intrinsic robustness limit for a given classification task.
>
> __Ways to improve adversarial attacks and defenses:__ A higher label uncertainty suggests the given image is more difficult to classify correctly. Thus, if an attacker has access to the label uncertainty information for a set of candidate images, then it will be preferable to attack those images with high label uncertainty from an attacker perspective. On the other hand, the notion of label uncertainty is defined intrinsically with respect to human classification, which suggests that we should treat the set of examples with high label uncertainty differently in a machine learning classifier, compared with examples with low label uncertainty. As we pointed out in Section 6.3, one potential way to improve adversarial defense is to add an abstaining option to the current state-of-the-art classifiers using estimates of label uncertainty. Nevertheless, there exist challenges that require further study, such as whether label uncertainty can be effectively estimated, especially in an adversarial setting.
>
> __Need of soft labels:__ We admit that this is a current limitation of our framework. We regard extending our framework to datasets without human soft labels as an important future work, and are currently investigating other uncertainty quantification methods for more effective estimation of label uncertainty beyond confident learning.

---

> > ### Comment · Reviewer_jR8K · 2021-11-18
> > **Response to Paper3096 Authors**
> >
> > Dear authors:
> >
> > Thanks for your careful reply, which addresses my concerns well. The extension to datasets without human soft labels is a promising direction and deserves to be explored.
> >
> > Reviewer
> > jR8K

---

### Official Review · Reviewer_zzwx · 2021-11-02

**Correctness:** 3
**Technical Novelty And Significance:** 3
**Empirical Novelty And Significance:** 3
**Recommendation:** 6
**Confidence:** 4

**Main Review:**

**Novelty**:
The formulation and inclusion of label uncertainty are novel.

**Clarity**:
The paper is well written. Mathematical notations and definitions are clear and consistent. The majority of the technical content is clear. However, a few points remain unclear.
1) Figure 4 captions are not clear. I assume that in figure 4 (a), examples are sorted in ascending label uncertainty order, and in figure 4 (b), examples are sorted in descending label uncertainty order.
2) In multiple places, the authors mention *robust accuracy*. However, there lacks any formal definition of this term.
3) Why is *empirical measure* denoted as *empirical risk* in figure 3 and several other places? I would appreciate it if the authors could elaborate on the relationship between the empirical measure and empirical risk. It seems that they are equivalent numerically.

**Technical questions**:
While I think the paper is mathematically rigorous, there are some confusing descriptions that hinder understanding.
1) This work claims to achieve a more accurate robustness estimate by deriving an *upper bound* on the intrinsic robustness, and in figure 3, the proposed method demonstrates uniformly lower values compared to the baseline. Confusingly, prior work [1] also claims to achieve a more accurate estimate. However, prior work [1] derived a *lower bound*. It is not clear which work provides a better estimate since there is no direct discussion on the importance of upper and lower bounds and no direct empirical comparison. Overall, I am confused about what constitutes a *better* estimate. This is also related to the next question.
2) in experiment section 6.2, the authors claim that being closer to the robust accuracy indicates a more accurate characterization of intrinsic robustness. It is not clear why this is the case because there is no discussion on the relationship between intrinsic robustness and robust accuracy. This seems to be the core statement supporting the paper's claim that including label uncertainty indeed improves estimation.

**Significance**: This line of works focuses on empirically estimating the intrinsic robustness of a dataset. Prior works [1] conclude that concentration of measure is not the main contributor to adversarial vulnerability. It is not clear what conclusions we can draw from this paper regarding the source of adversarial vulnerability.

**Other concerns**:
1) The paper lists many prior works. However, there is no direct comparison with them other than a baseline.  While there is a theoretical difference, i.e., prior works do not include label uncertainty, it is not clear if they produce less accurate robustness estimates.

2) Abstaining from uncertain predictions seems to be a rather indirect way to address the robustness issue. It is equivalent to counting only confident predictions in order to improve accuracy.

**Relevant Discussion**:
The paper proposes a new formulation of label uncertainty. It seems that the concept of label uncertainty also relates to the concept of *aleatoric uncertainty* [2] that captures the inherent ambiguity of label assignment. Is it possible to extend the current framework to include other definitions of label uncertainty?

[1] Prescott, Jack, Xiao Zhang, and David Evans. "Improved Estimation of Concentration Under $\ell_p $-Norm Distance Metrics Using Half Spaces." arXiv preprint arXiv:2103.12913 (2021).

[2] Hüllermeier, Eyke, and Willem Waegeman. "Aleatoric and epistemic uncertainty in machine learning: An introduction to concepts and methods." Machine Learning 110.3 (2021): 457-506.

**Summary Of The Paper:**

The paper proposes to include label uncertainty (LU) in the formulation of the concentration of measure problem which has been deemed the cause of adversarial vulnerability. It suggests that the current formulation of intrinsic robustness based on concentration measures is insufficient because of the exclusion of label information. The paper demonstrates a more accurate estimation of intrinsic robustness. Practically, in the original concentration of measure problem, a search algorithm finds the least $\epsilon$-expansion set $\mu(\varepsilon_\epsilon)$ given a concentration constraint $\mu(\varepsilon)>\alpha$ from a selected collection of data sets. The new formulation adds a label uncertainty constraint to this search algorithm: the label uncertainty of the set $\varepsilon$ must be larger than $\gamma$.

**Summary Of The Review:**

The paper is mathematically rigorous and well-motivated. However, some unclear descriptions hinder understanding of the paper fully and it lacks direct empirical comparisons to prior works. I would be happy to raise my score if the authors can address my questions.

---

> ### Author Response · Authors · 2021-11-16
> **Response to Reviewer zzwx Regarding the technical questions - Part 1**
>
> Thank you for the detailed and insightful comments. We provide clarifications on the two technical questions, which are copied below. (We have also uploaded an updated paper with a new version of Figure 3 to address Q1, explained below.)
>
> Q1: “This work claims to achieve a more accurate robustness estimate by deriving an upper bound on the intrinsic robustness, and in figure 3, the proposed method demonstrates uniformly lower values compared to the baseline. Confusingly, prior work [1] also claims to achieve a more accurate estimate. However, prior work [1] derived a lower bound. It is not clear which work provides a better estimate since there is no direct discussion on the importance of upper and lower bounds and no direct empirical comparison. Overall, I am confused about what constitutes a better estimate. This is also related to the next question.”
>
> Q2: “In experiment section 6.2, the authors claim that being closer to the robust accuracy indicates a more accurate characterization of intrinsic robustness. It is not clear why this is the case because there is no discussion on the relationship between intrinsic robustness and robust accuracy. This seems to be the core statement supporting the paper's claim that including label uncertainty indeed improves estimation.”
>
> We start with Q2, then will address Q1.
> ------------------------------
> First, we clarify the relationship between _intrinsic robustness_ and _robust accuracy_. Robust accuracy stands for an empirical approximation of adversarial robustness (Definition 2.1), which is usually computed on a set of testing examples based on a certain type of attack. In particular, the robust accuracies of the RobustBench models used in our experiments are evaluated based on _AutoAttack_, as adopted by Croce et al. (2020). On the other hand, intrinsic robustness is defined as the maximum adversarial robustness with respect to some collection of classifiers $\mathcal{F}$ (Definition 2.2). Therefore, as long as robust accuracy is a good empirical approximation of adversarial robustness, the intrinsic robustness with respect to $\mathcal{F}$ can be understood as an upper bound on the robust accuracy of any classifier $f\in\mathcal{F}$.
>
> Note that the intrinsic robustness estimation requires specifying the collection of classifiers $\mathcal{F}$ under consideration. Previous works (Mahloujifar et al., 2019b; Prescott et al., 2021) choose $\mathcal{F}$ to be the set of imperfect classifiers $\mathcal{F}_\alpha$ with risk at least $\alpha$. The reason for such choice is two-fold: (1) intrinsic robustness defined with $\mathcal{F}_\alpha$ connects with the standard concentration of measure, thus one can leverage techniques for measuring concentration to empirically estimate such intrinsic robustness; (2) $\mathcal{F}_\alpha$ is general enough such that it will cover any classification model learned for the given task, thus intrinsic robustness defined with $\mathcal{F}_\alpha$ can be viewed as a model-independent robustness upper bound, inherent to the underlying robust classification task.
>
> However, a fundamental question raised in this work is whether defining intrinsic robustness in terms of $\mathcal{F}_\alpha$ is meaningful, or more concretely, whether it reflects the inherent difficulty of developing robust classifiers for the underlying robust classification task. As discussed in Section 3 of our paper, intrinsic robustness defined with $\mathcal{F}_\alpha$ ignores data labels, an important component for learning an adversarially robust classifier, thus we argue that it fails to capture a faithful intrinsic limit for the underlying task. The driving goal of our work is to understand the intrinsic limit better by incorporating label uncertainty.
>
> Next, we clarify why imposing an additional label uncertainty constraint on $\mathcal{F}_\alpha$ yields a more accurate characterization of intrinsic robustness. What we mean by “intrinsic robustness” here can be understood as the maximum adversarial robustness one can hope to achieve by an optimal classifier produced using some (supervised) learning method. We expect such a notion of intrinsic robustness reveals the most accurate characterization of the adversarial vulnerability that can be attributed to the underlying robust classification task. Ideally, an estimate of intrinsic robustness with respect to the set of learnable classifiers is most desirable. However, rigorously defining the set of learnable classifiers is difficult, thus we propose to study the behavior of state-of-the-art (adversarially-trained) classification models as instances of “near-optimal” classifiers we are able to obtain.

---

> > ### Author Response · Authors · 2021-11-16
> > **Response to Reviewer zzwx Regarding the technical questions - Part 2**
> >
> > Observing the strong correlation between risk and label uncertainty, as well as the tendency to misclassify examples with higher label uncertainty by state-of-the-art classifiers (Figure 2), we hypothesize that a large error region label uncertainty should be a property that is shared by any predictive classifier produced by a learning method. As long as this hypothesis holds true, imposing a proper constraint on error region label uncertainty reduces the set size of $\mathcal{F_\alpha}$, while covering the unknown optimally-learned classifier, thus is expected to produce a tighter upper bound on intrinsic robustness. This is empirically confirmed by the experimental results presented in Section 6.2 of our paper, as our estimates of intrinsic robustness with label uncertainty constraint are closer to the robust accuracies achieved by state-of-the-art RobustBench models.
> >
> > Now we turn to address Q1.
> > ------------------------------
> > As explained in our response to Q2, we aim to obtain a more accurate understanding of the adversarial vulnerability intrinsic to the property of the underlying robust classification task, since we identify that characterizing the intrinsic robustness with respect to $\mathcal{F_\alpha}$ is not sufficient. Therefore, our work is different from Prescott et al. (2021) in terms of the research goal, as they aim to improve the empirical estimation of intrinsic robustness defined with $\mathcal{F_\alpha}$. The reason for claiming a lower bound in Prescott et al. (2021) is that the proposed method finds the best subset for the empirical concentration problem by restricting the search space to the set of half-spaces, thus when translated into an empirical estimate of intrinsic robustness, it becomes a lower bound. Restricting the whole search space into a smaller manageable subset is a necessary step for empirically measuring concentration, which was adopted in Mahloujifar et al. (2019b) and our work as well. In particular, Prescott et al. (2021) showed that restricting the search space to the set of half-spaces yields tighter empirical estimates than the choice specified in Mahloujifar et al. (2019b).
> >
> > For better illustration, we conduct additional experiments to compare our method with Prescott et al. (2021) when label uncertainty constraint is not involved, and update Figure 3 in our paper accordingly. The results show that the proposed empirical method in Prescott et al. (2021) is able to find tighter estimates for intrinsic robustness with respect to $\mathcal{F_\alpha}$, especially under $\ell_\infty$ perturbation. Note that this does not contradict with our claim that understanding intrinsic robustness defined with $\mathcal{F_{\alpha,\gamma}}$ is more meaningful. However, one may raise a question of whether the method proposed in Prescott et al. (2021) can be adapted to produce better empirical estimates under the setting, where a label uncertainty constraint is imposed. We made an initial attempt for this by modifying the algorithm of Prescott et al. (2021) to select the half-space whose label uncertainty is at least $\gamma=0.17$, but it turns out that it is not able to return a feasible half-space satisfying such constraint on CIFAR-10. In contrast, our proposed method is based on the union of $\ell_p$-balls, which is a universal approximator of arbitrary set, thus is more flexible in finding feasible sets within $\mathcal{F_{\alpha,\gamma}}$.
> >
> > Finally, in terms of “what constitutes a better estimate”, we should first define a good “upper bound” by choosing a meaningful $\mathcal{F}$ for defining intrinsic robustness, such that it covers the set of classification models of interest and is as close as possible. We should also try to produce a better “lower bound” by finding the right empirical method, such that it produces a tight estimate of intrinsic robustness with respect to the specified $\mathcal{F}$. The two pursuits are not contradictory to each other, but the first pursuit should come before the second one. While the main focus of our paper is to show that intrinsic robustness defined with $\mathcal{F_{\alpha,\gamma}}$ is more meaningful, it would be an important next step to study how tight our empirical estimate is for approximating the intrinsic robustness with respect to $\mathcal{F_{\alpha,\gamma}}$.

---

> > > ### Author Response · Authors · 2021-11-16
> > > **Response to other comments:**
> > >
> > > __Figure 4 captions:__ We have updated the captions of Figure 4 according to your comment.
> > >
> > > __Definition of robust accuracy:__ Robust accuracy stands for empirical adversarial risk computed on a set of testing examples based on a certain type of attack. In our experiments, the robust accuracies of RobustBench models are evaluated based on AutoAttack (see Appendix G of our paper).
> > >
> > > __Connection between empirical risk and empirical measure:__ According to Definition 2.1, the risk of any classifier $f$ is equivalent to the measure $\mu$ of its induced error region with respect to the concept function $c$. To be more specific, if we denote the error region of $f$ as $\mathcal{E}_f$, $\mathrm{Risk}(f,c) = \mu(\mathcal{E}_f)$ holds for any classifier $f$. Such equivalence also applies to the empirical case where we only have access to a set of inputs sampled from $\mu$. Similarly, for any input region $\mathcal{E}$, we can also construct a classifier $f$, where $f(x) = c(x)$ if $x\not\in\mathcal{E}$ and $f(x) \neq c(x)$ otherwise, such that the risk of $f$ matches the $\mu(\mathcal{E}).$
> > >
> > > __Extension to other notions of uncertainty:__ Thank you for raising this relevant discussion between our definition of label uncertainty and other notions of uncertainty, such as aleatoric and epistemic uncertainty.
> > >
> > > To the best of our knowledge, _aleatoric uncertainty_ stands for the irreducible uncertainty inherent to the data distribution. For classification tasks, aleatoric uncertainty arises because of class overlap. This type of uncertainty can be characterized by human soft labels, where inputs with high aleatoric uncertainty should correspond to inputs whose class label human annotators largely disagree with. Therefore, our concentration estimation framework can be directly extended to study the effect of aleatoric uncertainty on intrinsic robustness. Note that the only difference between our definition of label uncertainty and aleatoric uncertainty is that the original label assignment of the concept function is considered in our definition, whereas aleatoric uncertainty usually assumes class labels are generated according to the underlying conditional distribution $P(Y|X)$, thus concept function is not involved.
> > >
> > > In addition, _epistemic uncertainty_ occurs due to inadequate knowledge and data, which is reducible if we have an infinite amount of data and the knowledge of the optimal learning method. Note that we aim to characterize a model-independent intrinsic limit, which essentially corresponds to the optimal robust classifier one hopes to learn for the given task, thus our framework can not be applied to account for epistemic uncertainty. Nevertheless, for a specific learning method, understanding the adversarial vulnerability that is caused by epistemic uncertainty would be an interesting future work.

---

### Official Review · Reviewer_HNrN · 2021-11-02

**Correctness:** 4
**Technical Novelty And Significance:** 4
**Empirical Novelty And Significance:** 4
**Recommendation:** 8
**Confidence:** 2

**Main Review:**

The paper is clear and well written. The main concepts and motivation are clearly introduced and described. Discussion is also clear and informative, showing the relations of label uncertainty with model classification errors.


**Summary Of The Paper:**

The paper discusses model robustness, claiming that the current studies on the concentration of measurement do not consider label uncertainty. Hence the presented work defines the concept of label uncertainty and suggests a concentration of measurement that incorporates this uncertainty to bring a more accurate model robustness assessment.


**Summary Of The Review:**

The paper, contribution, and discussion are clear and well written. It is an interesting paper.

---

### Official Review · Reviewer_rWcj · 2021-11-02

**Correctness:** 4
**Technical Novelty And Significance:** 3
**Empirical Novelty And Significance:** 3
**Recommendation:** 6
**Confidence:** 4

**Main Review:**

- The paper is clearly written. Especially, I liked that the paper carefully delivers the preliminaries and adequately covers the literature. I also feel the theory and method are well-motivated, and addresses an important problem of estimating the upper bound of adversarial robustness achievable by the current state-of-the-art classifiers. The Gaussian mixture model example given in Section 3 clearly presents the motivation of the method.
- One may still concern, however, that the theory (and the empirical method from it) is largely based on Mahloujifar et al. (2019) with some incremental points of considering label uncertainty upon it. Nevertheless, I still feel the work is worth to be shared to the community, given that incorporating label uncertainty for intrinsic robustness is yet a novel aspect, suggesting some useful insights not only for the context of adversarial robustness, but for, e.g., confidence calibration or label noises.
- I also think the experimental results can be a weakness of the paper. Although the paper confirms their major claims focusing on CIFAR-10H, one may concern that there is no straightforward way to extend these results into other datasets, especially when there is no uncertainty information available. This can reduce the technical significance of the proposed method. The paper could present some results on other datasets with an approximative label uncertanty, e.g., as done in Appendix E on CIFAR-10.

**Summary Of The Paper:**

The paper improves the previous results on the intrinsic robustness (i.e., an upper bound on the adversarial robustness over a set of classifiers) based on concentration of data distribution, by incorporating the constraint on the label uncertainty of the models. Specifically, the paper tightens the upper bound on the adversarial robustness given a family of models with error rates $\ge \alpha$, by additionally assuming that the average label uncertainty must be $\ge \gamma$. This requires an information of label uncertainty for each data sample, so the paper leveraged CIFAR-10H (CIFAR-10 with human-annotated uncertainty) in their experiments. Experimental results on CIFAR-10/CIFAR-10H show that the proposed upper bound can be practically computable, and improves the previous upper bound given the knowledge of CIFAR-10H.

**Summary Of The Review:**

Overall, I found the paper is clearly written, well-motivated, and addresses an important problem. Although I feel one may concern on the technical contribution and experimental results, I still feel the work is worth to be shared to the community, given that incorporating label uncertainty for intrinsic robustness is yet a novel aspect, suggesting some useful insights not only for the context of adversarial robustness, but for, e.g., confidence calibration or label noises.

---

> ### Author Response · Authors · 2021-11-16
> **Response to Reviewer rWcj**
>
> Thank you for recognizing the novelty of our work, and for the encouraging feedback. Below, we respond to the concerns raised by the reviewer.
>
> __The theory (and the empirical method from it) is largely based on Mahloujifar et al. (2019):__ Yes, our theory and method build upon that work, the novelty and importance of our work lies in the introduction of label uncertainty. We argue that this is essential to understanding intrinsic robustness, develop new formal definitions and theoretical results to support label uncertainty, and extend the empirical method from Mahloujifar et al. (2019) to incorporate label uncertainty and demonstrate how it improves the results.
>
> __Extension to other datasets, when there is no uncertainty information available:__ As we discuss in Section 4, we acknowledge that the requirement of label uncertainty information is a limitation of our work. However, since our experiments on CIFAR-10H reveal an obvious misalignment between human recognized errors and errors produced by confident learning (see Appendix E), we are not inclined towards extending our concentration estimation framework to other datasets using such label uncertainty approximation techniques. We think the first priority is to understand the reason behind such misalignment and study whether there are better label uncertainty approximation methods. We view understanding the intrinsic robustness limit for dataset without human soft labels as an important next step of our work.

---

### Author Response · Authors · 2021-11-16
**An updated version of our paper**

We thank all the reviewers for putting the time and effort in reviewing our work and making the manuscript better. We have uploaded an updated paper with a new version of Figure 3, including additional baseline estimates of intrinsic robustness produced using the empirical method proposed by Prescott et al. (2021). In addition, we updated the captions and discussions of Figure 3 in Section 6.2, and updated Figure 4 captions in Section 6.3 for a better presentation. All these updates are highlighted in blue.

---

### Decision · Program_Chairs · 2022-01-20

**Decision:**

Accept (Poster)

**Comment:**

The paper was praised for being clearly written, well-motivated, and for addressing an important problem: measuring intrinsic robustness.
It improves the previous results on intrinsic robustness based on concentration of data distribution, by incorporating the constraint on the label uncertainty of the models.
This requires information on label uncertainty for each data sample rarely available (here CIFAR-10H is considered), but could open new directions for future work on adversarial robustness, confidence calibration or label noises.